# TP53 mutations and RNA-binding protein MUSASHI-2 drive resistance to PRMT5-targeted therapy in B-cell lymphoma

Tatiana Erazo[1], Chiara M. Evans[1,2], Daniel Zakheim[3], Eren L. Chu[1], Alice Yunsi Refermat [3], Zahra Asgari[4], Xuejing Yang[1], Mariana Da Silva Ferreira[1], Sanjoy Mehta[3], Marco Vincenzo Russo[3], Andrea Knezevic[5], Xi-Ping Zhang[6], Zhengming Chen[7], Myles Fennell[3], Ralph Garippa[3], Venkatraman Seshan [5], Elisa de Stanchina[8], Olena Barbash[6], Connie Lee Batlevi [4], Christina S. Leslie[9], Ari M. Melnick [10], Anas Younes [4,11] ✉ & Michael G. Kharas [1,11] ✉

To identify drivers of sensitivity and resistance to Protein Arginine Methyl-transferase 5 (PRMT5) inhibition, we perform a genome-wide CRISPR/Cas9 screen. We identify *TP53* and RNA-binding protein *MUSASHI2* (*MSI2*) as the top-ranked sensitizer and driver of resistance to specific PRMT5i, GSK-591, respectively. *TP53* deletion and *TP53*[R248W] mutation are biomarkers of resistance to GSK-591. *PRMT5* expression correlates with *MSI2* expression in lymphoma patients. MSI2 depletion and pharmacological inhibition using Ro 08-2750 (Ro) both synergize with GSK-591 to reduce cell growth. Ro reduces MSI2 binding to its global targets and dual treatment of Ro and PRMT5 inhibitors result in synergistic gene expression changes including cell cycle, P53 and MYC signatures. Dual MSI2 and PRMT5 inhibition further blocks c-MYC and BCL-2 translation. BCL-2 depletion or inhibition with venetoclax synergizes with a PRMT5 inhibitor by inducing reduced cell growth and apoptosis. Thus, we propose a therapeutic strategy in lymphoma that combines PRMT5 with MSI2 or BCL-2 inhibition.

The protein arginine methyltransferases (PRMTs) family is a group of nine enzymes that transfer a methyl group from a cofactor, S-adenosyl methionine, to arginine residues on their target protein. PRMTs are classified into three groups based on their methylation pattern: monomethylation, symmetric dimethylation, or asymmetric dimethylation[1]. Protein arginine methyltransferase-5 (PRMT5) is the major enzyme responsible for the symmetric dimethylation of arginine on a variety of cytoplasmic and nuclear substrates, including histones, most notably at H3R2, H3R8, H4R3, and H2AR3, and these can result in both repression or transcription of genes[2]. PRMT5 knockout mice are embryonically lethal and conditional knockout studies in mice demonstrate major defects in the function of hematopoietic stem cells and B cells in various stages of development[3,4].

Non-Hodgkin's lymphoma (NHL) is the 7th most common cancer in adults and remains largely incurable with current treatment modalities. PRMT5 is overexpressed in germinal center-derived lymphomas

[1]Molecular Pharmacology Program, Experimental Therapeutics Center and Center for Stem Cell Biology, Memorial Sloan Kettering Cancer Center, New York, NY, USA. [2]Department of Pharmacology, Weill Cornell School of Medical Sciences, New York, NY, USA. [3]Gene Editing and Screening Core, Memorial Sloan Kettering Cancer Center, New York, NY, USA. [4]Lymphoma Service, Memorial Sloan Kettering Cancer Center, New York, NY, USA. [5]Department of Epidemiology and Biostatistics, Memorial Sloan Kettering Cancer Center, New York, NY, USA. [6]Epigenetics Research Unit, GlaxoSmithKline, Collegeville, PA 19426, USA. [7]Division of Biostatistics and Epidemiology, Weill Cornell Medicine, New York, NY 10021, USA. [8]Antitumor Assessment Core, Memorial Sloan Kettering Cancer Center, New York, NY, USA. [9]Computational Biology Program, Memorial Sloan Kettering Cancer Center, New York, NY, USA. [10]Division of Hematology and Medical Oncology, Sanford I. Weill Department of Medicine, Weill Cornell Medicine, New York, NY, USA. [11]These authors jointly supervised this work: Anas Younes, Michael G. Kharas. ✉e-mail: anas.younes@astrazeneca.com; kharasm@mskcc.org

and mantle cell lymphoma and plays an important role in lymphomagenesis by regulating several oncogenic pathways[5–8]. Among others, PRMT5 methylates P53 thereby modifying its DNA-binding activity and suppressing the expression of proapoptotic and antiproliferative genes, which contributes to lymphomagenesis[7,9]. Furthermore, PRMT5 increases de novo syntheses of key lymphoma-drivers, MYC and CYCLIN D1 by repressing miR-33b, miR-96, and miR-503[8,10]. Moreover, PRMT5 cooperates with MYC to sustain splicing fidelity, which is key to ensure tumor maintenance of MYC-driven lymphomas[11], and promotes the upregulation of BCL-6 repressive activity in DLBCL[12]. Collectively, these data suggest selective targeting of PRMT5 may have a therapeutic value in lymphoma. This led to the development of the first selective inhibitors of PRMT5, of which three have progressed into clinical trials.

Here, we study the mechanisms of resistance and sensitivity to the selective PRMT5 inhibitor, GSK3203591 (GSK-591). This compound belongs to a series of non-SAM analogs that inhibit PRMT5 by occupying the substrate binding pocket in the presence of the PRMT5's cofactors, MEP50 and SAM[13]. Currently, its close analogue, GSK3326595, was the first to be approved for Phase I clinical trials in patients with NHL and advanced solid tumors (NCT02783300). Due to the increased interest of PRMT5-targeted therapy application in the clinical setting and the potential development of drug resistance, we investigated the mechanisms of resistance to PRMT5 inhibition. To this end, we performed a genome-wide CRISPR/Cas9 screen in the mantle cell lymphoma line, Z-138 and a validation CRISPR/Cas9 screen in the DLBCL cell line, OCI-LY19. The top sensitizing gene was *TP53*, which validated our screens, as it is a well characterized target of PRMT5[7,9]. Furthermore, we found that *TP53* deletion and the hot spot *TP53*[R248W] mutation are biomarkers of resistance to PRMT5-targeted therapy.

We identified the RNA-binding protein MUSASHI-2 (MSI2), as the top driver of resistance to PRMT5 inhibition. MSI2 controls protein translation by binding primarily to the 3′UTRs of mRNAs. MSI2 plays a key role in hematopoietic stem cell activation, myeloid leukemia, and chronic lymphocytic leukemia (CLL) and its expression correlates with poor prognosis in multiple hematological malignancies, but its role in B-cell lymphoma is not currently known[14–19]. We found that c-MYC and BCL-2 are key targets of the MSI2/PRMT5 axis, which drives resistance to PRMT5 inhibition in B-cell lymphomas.

Moreover, our study provides mechanism-based combination strategies to overcome resistance to PRMT5 inhibitors that can define therapeutic interventions to be applied in further clinical trials. First, the combination of GSK-591 with Ro 08-2750, which inhibits MSI2 activity, results in loss of c-MYC and BCL-2 protein expression. The second strategy utilizes dual targeting of PRMT5 and BCL-2, using GSK-591 and venetoclax, that exhibits potent anti-tumor efficacy and induction of apoptosis across a wide range of lymphomas.

## Results

### CRISPR/Cas9 KO screens to identify drivers of resistance and sensitivity to PRMT5 inhibition

Previous studies have shown a role for PRMT5 in lymphoma[5,12,20,21], and to expand on these studies, we induced genetic depletion of PRMT5 with a short hairpin RNA (shRNA). PRMT5 knockdown reduced proliferation in 6 cell lines across different subtypes of B-cell lymphoma (Fig. 1A). Although three PRMT5 inhibitors are being evaluated clinically, we focused our study on GSK-591, which is a close analogue of the first selective PRMT5 inhibitor (GSK3326595) approved for human studies. First, we assessed the antiproliferative activity of GSK-591, in a panel of 31 lymphoma cell lines, representing a wide range of histologic subtypes. GSK-591 inhibited cell proliferation across all lymphoma subtypes. Seventeen (55%) had an IC50 of < 5 μM, and 14 (45%) lymphoma cell lines were less sensitive to GSK-591, with an IC50 > 5μM (Fig. 1B). In all cell lines, GSK-591 reduced PRMT5 enzymatic activity, as evident by decreasing arginine

dimethylation of Histone 3, SmD3, and Histone 4 without affecting PRMT5 or MEP50 (Supplementary Fig. 1A). Reintroducing and over-expressing the wild type PRMT5 but not the catalytic dead double mutant PRMT5 (PRMT5[G367A/R368A]) rescues Z-138 cells from PRMT5 knockdown-mediated toxicity (Supplementary Fig. 1B, C). These results demonstrate that PRMT5 expression and its enzymatic activity are required for lymphoma cell survival and confirm the potential role of PRMT5 as a therapeutic target in lymphoma.

In vivo studies with GSK-591 are limited due to its poor pharmacological properties[13], we evaluated the in vivo efficacy of PRMT5 inhibition using the close analog GSK-025, because of its equivalent mechanism of action and cytotoxic activity in vitro in all lymphoma cell lines tested[22]. PRMT5 inhibition significantly reduced tumor volume in an MCL and a DLBCL patient-derived xenograft and Z-138 xenograft models without inducing toxicity, as shown by the maintenance of the animal's weight (Supplementary Fig. 1D).

We hypothesized that the resistance to GSK-591 observed in some lymphoma cell lines and the modest effect of PRMT5 inhibitor in vivo are likely related to the induction of feedback survival mechanisms and/or due to the coexistence of parallel survival pathways. Thus, in order to identify these mechanisms, we performed a genome-wide CRISPR/Cas9 knockout screen. To this end, the mantle cell lymphoma cell line, Z-138 with constitutively expressed Cas9, was transduced with the human lentiviral Brunello library targeting the entire human genome, at an MOI of 0.3. To select for GSK591-resistant Z-138 cells, we treated the cells transduced with the Brunello library with 0.5 uM of GSK-591, which is 20-fold higher than its IC50, for 8 days. We extracted genomic DNA from Brunello-transduced cells, as a control for the screen (T0) and from DMSO and GSK591-treated cells, then the sgRNA abundance was measured by deep sequencing (Fig. 1C). Normalized sgRNA counts revealed a near complete library representation at Day 0 (T0) and after 8 days of treatment with DMSO or GSK-591 (Supplementary Fig. 1E). Pearson correlation analysis demonstrates a high correlation between time zero control (T0), DMSO and GSK591-treated samples (Supplementary Fig. 1F). These data demonstrate that the drug treatment did not induce high depletion of sgRNA in the screen.

For the data analysis, we first identified the genetic vulnerabilities in lymphoma cells. Of the 19,114 genes targeted by the Brunello library, 85 scored (FDR < 0.05 and logFC < −2) as essential genes for lymphoma cells survival in our unbiased CRISPR screen (Supplementary Fig. 1G and Supplementary Data 1). To identify the genes that drive resistance and sensitivity to GSK-591 activity, we first removed from the downstream functional analysis the essential genes identified in Supplementary Fig. 1G. Then, using the CAMERA (Correlation Adjusted MEan RAnk) function in edgeR, we generated the top ranking sgRNAs (adjust *p*-value < 0.1 and log2FC < −2 for depletion and log2FC > 2 for enrichment), which yielded 316 sensitizing genes and 89 genes involved in resistance to GSK-591 (Fig. 1D and Supplementary Data 2).

To validate the results from our genome-wide CRISPR screen, we designed a focused CRISPR/Cas9 knockout library containing 2020 sgRNAs targeting the top 405 sensitizing and resistant genes, 85 essential genes, and ~400 negative control sgRNAs. For the validation screen, we used the DLBCL cell line, OCI-LY19 with stably expressed Cas9. A similar screening strategy was applied as in the whole-genome CRISPR screen, and deep sequencing data was used to rank the genes using the CAMERA method. This validation CRISPR screen identified 32 sensitizing genes and 14 genes involved in resistance to GSK-591 (adjusted *p*-value < 0.1 and log2FC < −2 and log2FC > 2) (Fig. 1E and Supplementary Data 3).

To identify the pathways required for lymphoma cell growth and survival, we performed gene enrichment analysis using ClusterProfiler. As expected, genes implicated in essential processes such as mito-chondrial and ribosomal biogenesis were highly enriched in the

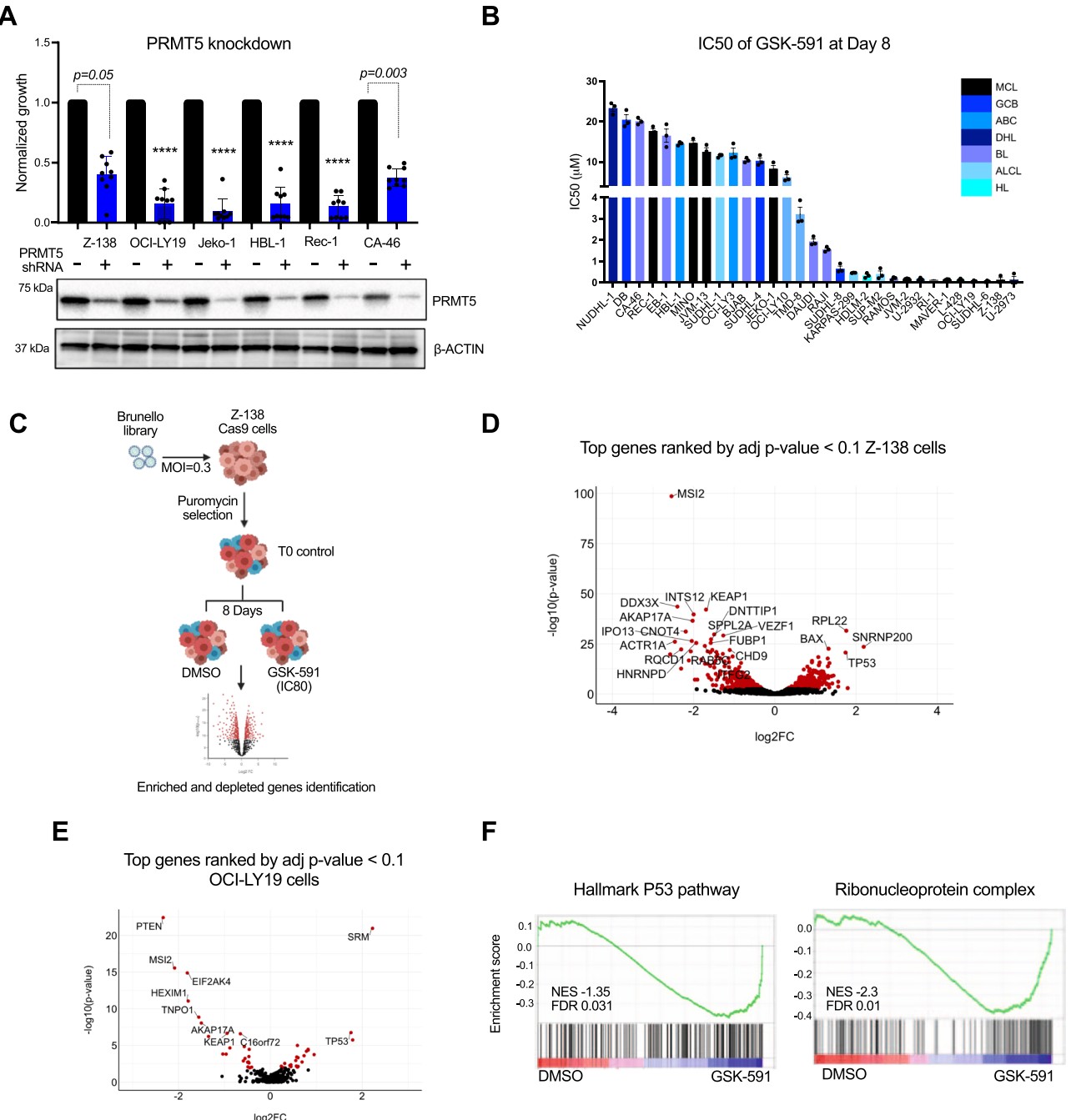

**Fig. 1 | Genome-wide CRISPR/Cas9 KO screen identified controllers of sensitivity and resistance to PRMT5 inhibition. A** Lymphoma cells were cells transduced with lentivirus expressing shRNA targeting PRMT5 or scramble. Cell viability and immunoblot analysis were assessed 5 days after transduction and puromycin selection. Data represent mean ± SEM of $n$ = 3 replicates examined over 3 independent experiments. $p$-values are indicated, ****$p$ < 0.0001, statistical analysis by two-sided Student's $t$ test. **B** Bar graph showing IC50 values of GSK-591 after 8 days of treatment in 31 lymphoma cell lines from different lymphoma subtypes: Mantle cell lymphoma, Germinal center DLBCL (GCB), Activated B-cell DLBCL (ABC), Double hit lymphoma (DHL), Burkitt's lymphoma (BL), Anaplastic large cell lymphoma (ALCL) and Hodgkin's lymphoma (HL). Cell viability was monitored by Celltiter-Glo assay. Error bars represent SEM of triplicate experiments. **C** Schematic representation of the GSK-591 CRISPR-Cas9 negative screening strategy in Z-138 Cas9 cells. **D** Volcano plot comparing significance of enrichment/depletion and

log2 fold change for all 4 sgRNAs per gene. Red dots represent the genes that were significant depleted and enriched defined by adjust $p$-value < 0.1 and log$_2$FC < −2 or log$_2$FC > 2. Top 20 hits are annotated. $p$-values and log$_2$FC were generated using the Wilcoxon-Mann-Whitney test in the CAMERA function. **E** Validation screen in OCI-LY19 cells: GSK-591 CRISPR-Cas9 negative screening was performed using a customized library (405 genes). Volcano plot showing significantly enriched and depleted genes (red dots). Top 10 hits are annotated. $p$-values and log$_2$FC were generated using the Wilcoxon-Mann-Whitney test in the CAMERA function. **F** GSEA analysis showing that P53 pathway and Ribonucleoprotein complex (MSI2 targets) are enriched among the genes that are differentially expressed in GSK-591-resistant cells compared to DMSO-treated cells. The black bars on the $x$-axis show the genes in the P53-pathway and MSI2-pathway CRISPR ranked list, with Log2fc (DSMO/GSK-591). NES normalized enrichment score. Source data are provided as a Source data file.

dropout genes. Pathway-level analysis also identified the S-adenosylmethionine-dependent methyltransferase pathway (Supplementary Fig. 1H and Supplementary Table T1). The overlap of the highly depleted genes in the T0 control between the two independent screens, confirmed that PRMT5 is one of the few genes identified in our CRISPR screens as essential for lymphoma and clinically actionable (Supplementary Fig. 1I). These findings validated the PRMT5 dependence in lymphoma that was previously shown to have therapeutic value in this disease.

To identify the genes specifically involved in GSK-591 activity, we overlapped the datasets from the two CRISPR screens and performed Gene Set Enrichment Analysis (GSEA) analysis. We found that *TP53* is the top significantly enriched pathway, and *TP53* as the top-ranked sensitizing hit to the PRMT5 inhibitor. The most significantly depleted genes belong to the ribonucleoprotein complex pathway (Supplementary Data 4). The RNA binding protein *MUSASHI-2* (*MSI2*) is found in this pathway and was identified as the top driver of resistance to GSK-591 (Fig. 1F). Consequently, we focused our target validation efforts on *TP53* and *MSI2*.

### *TP53* deletion and the hot spot *TP53^R248W^* mutation are biomarkers of resistance to PRMT5 inhibition

Our CRISPR screen data indicate that *TP53* may be a target of GSK-591. Accordingly, we investigated whether P53 wild type expression was essential for GSK-591 activity. Here, we used Z-138 and OCI-LY19 cells with wild type *TP53* and then generated isogenic, *TP53* stable knockout cells using CRISPR/Cas9 using sgRNAs targeting *TP53*. In the parental cells, GSK-591 treatment increased P53, PUMA, and BAX protein expression and induced PARP cleavage, but failed to do so in P53-deficient cells (Fig. 2A). Z-138 and OCI-LY19 parental *TP53*wt cells were sensitive to GSK-591, whereas, P53-deficient cells were resistant to the drug (Fig. 2B).

We reasoned if P53 is a major mediator of GSK-591 cytotoxicity, ectopic expression of wild type P53 should restore drug sensitivity in P53 mutant cells. To test this hypothesis, Mino cells that carry the mutation *TP53^V147G^* and Rec-1 cells carrying *TP53^G245D^* and *TP53^Q317*^* mutations were transfected with GFP-P53 wt and treated with GSK-591 for 72 h. Ectopically expressed P53 wt in combination with GSK-591 treatment resulted in upregulation of the P53 targets, P21, PUMA, BAX, and NOXA (Fig. 2C). Moreover, overexpression of P53 wt restored sensitivity to GSK-591 (Fig. 2D).

In a panel of lymphoma cell lines, we identified multiple *TP53* mutations at variable allele frequencies. Among the most prevalent in three of the GSK591-resistant cell lines (CA-46, DB, NUDHL-1) were biallelic mutations in R248 of *TP53* (Supplementary Table T2). The *TP53^R248W/Q^* is the single most common mutation in all *TP53*-altered tumor types accounting for 9% of cases which translates in the near 66,000 newly diagnosed cancer patients in the US per year[23]. Consistent with this, in primary lymphoma tissue specimens, *TP53* mutations were found in 20% of 1475 cases that were sequenced at MSKCC and *TP53^R248W/Q^* was the most frequent mutation in this cohort of lymphoma patients (Supplementary Fig. 2A, B).

To confirm the role of *TP53^R248W^* mutant in mediating resistance to GSK-591, we generated *TP53^R248W^* isogenic cells from Z-138 cells with CRISPR/Cas9 HDR-mediated targeting. PRMT5 inhibition increased the mRNA and protein abundance of P53 and its downstream targets BAX, P21, and MDM2 in the Z-138 cells carrying *TP53* wt gene, but it failed to induce these effects in isogenic cells that were either P53 deficient (P53 KO) or carrying *TP53^R248W^* (Fig. 2F and Supplementary Fig. 2C). *TP53^R248W^* cells were less sensitive to GSK-591 when compared with *TP53* wt cells (Fig. 2G). Consistent with the in vitro results, in vivo PRMT5 inhibition failed to prevent tumor progression in the *TP53^R248W^* xenograft model or in an MCL PDX (Fig. 2H-I). Collectively, our data suggest that *TP53^R248W^* mutation is a predictor of resistance to therapies utilizing PRMT5 inhibition.

### MSI2 is the top driver of resistance to PRMT5 inhibition

MSI2 plays a key role in myeloid and in chronic lymphoid leukemogenesis, however, its role in B-cell lymphomas remains unknown[14,16,18,19,24]. We analyzed the clinical relevance of MSI2 in lymphoma. *MSI2* mRNA expression was significantly upregulated in patients with DLBCL compared to normal B-cells (*p*-value < 0.01), (Fig. 3A). We examined the expression of *MSI2* across different DLBCL subtypes ABC, GCB, and other non-characterized subtypes from the BC Cancer Lymphoid Cancer Dataset (*n* = 322). We found that *MSI2* expression is significantly higher in GCB-DLBCL (Fig. 3B). Moreover, in a small DLBCL patients' dataset (TCGA; *n* = 47), higher expression of *MSI2* exhibited significantly shorter overall survival times compared to those patients with low expression of *MSI2* (Supplementary Fig. 3A). However, this finding could not be validated as an independent marker in larger datasets. We found a positive correlation of *PRMT5* and *MSI2* expression in patients with de novo ABC and GCB-DLBCL (*n* = 820), (Fig. 3C). Furthermore, MSI2 and PRMT5 protein abundance are lower in non-malignant B cells and de novo MCL and DLBCL primary samples compared to relapsed MCL, relapsed DLBCL, and lymphoma cell lines (Fig. 3D). To examine whether there is a link between the mechanisms of resistance to PRMT5 inhibitor driven by MSI2 and *TP53* mutations. We compared the IC50s of GSK-591 and MSI2 abundance determined by immunoblot and further subdivide the lymphoma cell lines based on *TP53* mutational status. We found that either high MSI2 abundance or *TP53* wt was significantly associated with lower IC50s values of GSK-591 (Fig. 3E). These data suggest that MSI2 and *TP53* are predictors of response to PRMT5 inhibitors.

We then modulated MSI2 expression to validate its role in driving resistance to PRMT5 inhibition. Forced MSI2 expression reversed sensitivity to the PRMT5 inhibitor treatment in Z-138 and OCI-LY19 cells across a range of concentrations to PRMT5 inhibitor (Fig. 3F and Supplementary Fig. 3B). Additionally, shRNA-mediated depletion of MSI2 modestly decreased lymphoma cell proliferation in two GSK591-sensitive cell lines, Z-138 and OCI-LY19, but resulted in a 10-fold decrease in the IC50 of GSK-591 (Fig. 3G). The GSK591-resistant cell lines, HBL-1 and CA-46 became sensitized to GSK-591 after MSI2 depletion (Supplementary Fig. 3C, D). Combination therapy demonstrated activity in vivo, since MSI2 depletion resulted in a 50% reduction of tumor growth which was further reduced with the addition of PRMT5 inhibitor (80% drop) without inducing toxicity (Fig. 3H and Supplementary Fig. 3E). We confirmed the synergistic effect of PRMT5 inhibition and MSI2 knockdown on cell proliferation by performing the immunohistochemical analysis of Ki-67, a biomarker of proliferating cancer cells[25] (Fig. 3I). The lymphoma cells resistant to the PRMT5 inhibitor were those that evaded successful MSI2 shRNA-mediated knockdown (Supplementary Fig. 3F). These data indicate that MSI2 contributes to the resistance to PRMT5 inhibition in vivo.

We then tested the effect of the pharmacological inhibition of MSI2 using Ro 08-2750 (Ro), a small molecule that inhibits MSI RNA-binding activity and has anti-proliferative activity in myeloid leukemia cells and chronic lymphocytic leukemia[19,26]. Consistent with our genetic studies, the combination of Ro and GSK-591 was synergistic, inducing greater anti-proliferative activity in lymphoma cells than each compound alone (Fig. 3J, K and Source Data). Furthermore, the dual treatment caused an increase in G2/M phase cells with a concomitant decrease in the number of cells in G1 phase, confirming that the combination of GSK-591 and Ro is synergistically inducing cell cycle arrest in B-cell lymphoma (Supplementary Fig. 3G). Both genetic and pharmacological approaches indicate that MSI2 and PRMT5 inhibition were synergistic in preventing growth of B-cell lymphoma cells.

### Combination of PRMT5 and MSI2 inhibitors induced global changes in the transcriptional program in lymphoma cells

To understand the molecular basis for the synergistic effect on cell viability and proliferation with dual MSI2 and PRMT5 inhibition, we

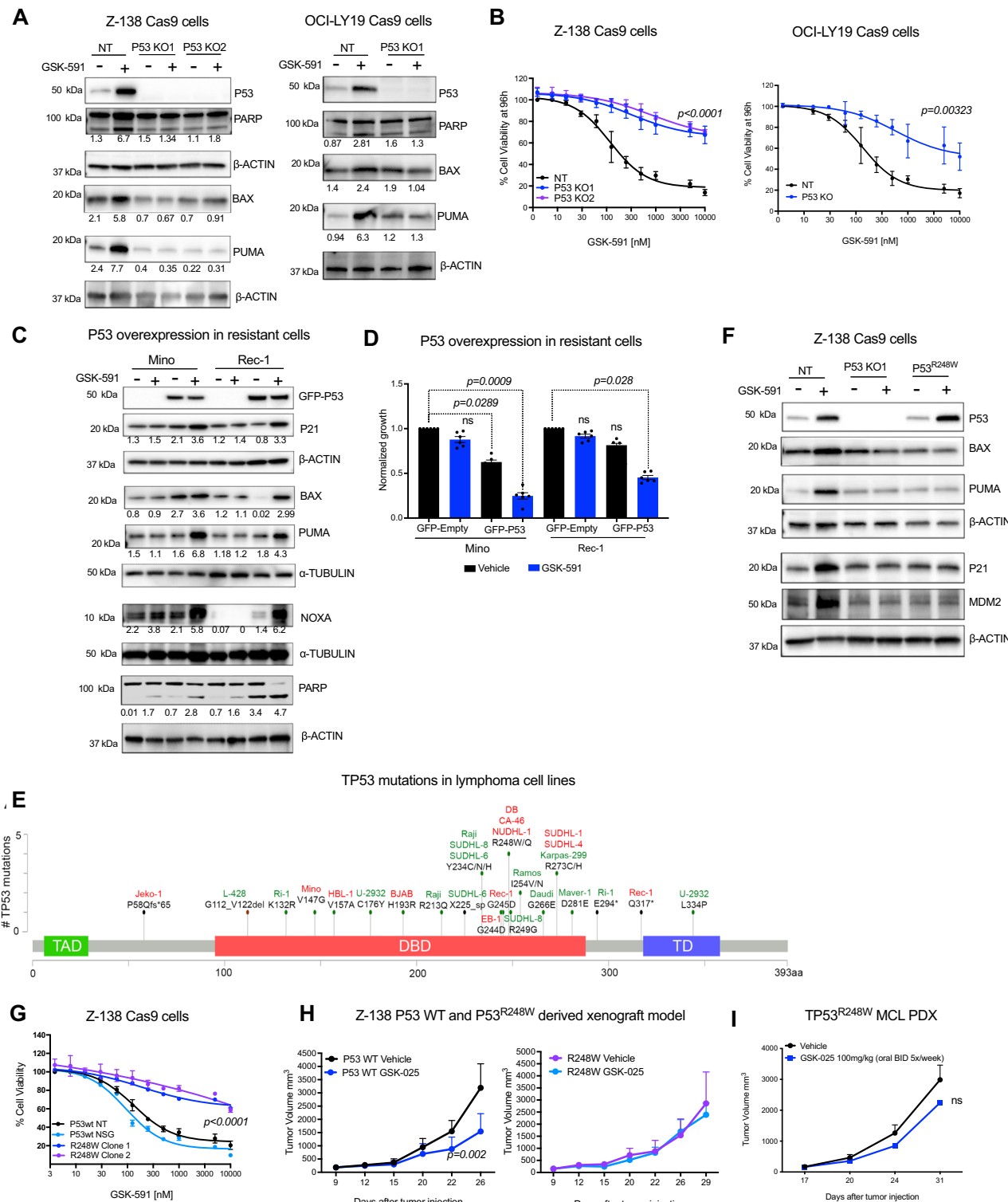

analyzed the gene expression changes. Surprisingly, treatment with the GSK-591 alone induced very few significant changes in the transcriptome, while Ro induced over 300 significant changes in transcripts (adjusted *p*-value < 0.05 and log2FC ≥2). However, the combination of Ro and GSK-591 synergistically induced changes in the transcriptome of Z-138 cells with 1531 transcripts upregulated and 1069 downregulated upon combination treatment (Fig. 4A,B and Supplementary Data 5). To identify the functional pathways that contribute to the drug synergy, we performed gene pathway analysis using

all gene sets in the Molecular Signatures Database (MSigDB) (Supplementary Data 6). Upregulated genes upon combination of PRMT5 and MSI2 inhibitors were significantly enriched for hypoxia and P53 pathway signatures (adjusted *p*-value < 0.01 and > 20 genes) (Fig. 4C, D, Supplementary Fig. 4A and Supplementary Data 7).

Among the downregulated genes, we found genes enriched in the cell cycle pathway, such as *CDK4, CDK2, CHK1, CDC25A,* and related transcription factors, including *E2F1* (adjusted *p*-value < 0.01 and > 20 genes) (Fig. 4E, Supplementary Fig. 4B). Interestingly, one of the most

**Fig. 2 | *TP53* deletion and *TP53*<sup>R248W</sup> mutation confer resistance to PRMT5 inhibition. A** Immunoblot showing efficient P53 knockout in Z-138 and OCI-LY19 Cas9 cells and the effect of GSK-591 (1 μM) for 72 h on the indicated proteins. **B** Cell viability assay showing that P53 knockout cells are resistant to GSK-591 compared to P53-expressing cells (NT; Non-targeting). Cell viability was assessed after 96 h of treatment. Error bars represent SEM of three different experiments. *p*-values are indicated, and calculated using a two-sided Student's *t* test. **C** Mino and Rec-1 cells (GSK-591-resistant) expressing GFP-P53 and treated with GSK-591 (5 μM). The effect on P53 pathway members was analyzed by immunoblot. **D** Mino and Rec-1 cells expressing GFP-P53 and 2 h after transfection were treated with GSK-591 (5 μM). Cell viability was monitored after 72 h. Data represent mean ± SEM of *n* = 3 replicates examined over 2 independent experiments. *pi* are indicated. **E** Lollipop plot showing *TP53* site-specific mutations in a panel of lymphoma cell lines. TP53 domains: TAD (trans-activation domain), DBD (DNA-binding domain) and TD (tetramerization domain). Sensitive cell lines in green: IC$_{50}$ < 5 μM and less sensitive in

red: IC$_{50}$ > 5 μM. **F** Representative immunoblot analysis of the effect of GSK-591 (1 μM) for 72 h on the expression of P53 pathway members in P53 parental, P53 KO1 and TP53<sup>R248W</sup> cells. Similar effects were observed in two independent experiments. **G** Effect of GSK-591 on viability of the indicated isogenic cell lines was measured 96 h after treatment. Error bars represent SEM of three different experiments. ****$p$ < 0.001, statistical analysis by one-way ANOVA. **H** Z-138 *TP53* wt (NT) and Z-138 *TP53*<sup>R248W</sup> (clone 2) were xenografted in NSG mice. Animals were treated with vehicle or GSK-025, 100 mg/kg twice/day for 21 days (*n* = 8/group). Data are represented as mean ± SEM. Differences among groups were calculated by two-sided ANOVA, *p*-value = 0.002. **I** Relapse MCL PDX line DFBL-44685 harboring *TP53*<sup>R248W</sup> mutation was xenografted subcutaneously in NSG mice. Mice (*n* = 8/group) were treated with vehicle or GSK-025, 100 mg/kg twice/day for 21 days. Tumor volume curves are shown. Data are represented as mean ± SEM. NS, Not significant, *p*-value = 0.8496, by two-sided ANOVA.

---

negatively enriched pathways upon drug combination is the MYC genes signature (Fig. 4F, G, Supplementary Fig. 4C and Supplementary Data 8), a key regulator of cell cycle progression[27]. We further interrogated the DeSeq data and determined the combination of GSK-591 and Ro induced significant changes on gene expression of MYC targets involved in cell cycle regulation (Fig. 4H). We further validated by immunoblot analysis the effect of GSK-591 and Ro on cell cycle regulators. Thus, we observed that combination of GSK-591 and Ro enhance the expression of P53 and its targets, P21 and MDM2, which is consistent with the dramatic decrease in cell cycle-related proteins, CYCLIN B1, CDK4, and RAD51 (Fig. 4I). Consistent with pharmacological inhibition, genetic depletion of both MSI2 and PRMT5 using shRNAs induced dramatic changes on cell cycle regulators' protein abundance (Supplementary Fig. 4D) and reduction of cell growth (Supplementary Fig. 4E). Taken together, PRMT5 and MSI2 cooperate in maintaining cell proliferation and the cell cycle program in B-cell lymphomas.

### c-MYC cooperates with MSI2 to drive resistance to PRMT5 inhibition

We then investigated how the PRMT5/MSI2 axis directly regulates c-MYC and its contribution to resistance to PRMT5 inhibition. c-MYC is an established essential regulator for B-cell lymphoma cell growth[28,29]. Moreover, we found previously that MSI2 binds and regulates c-MYC translation in myeloid leukemia cells[26,30]. We tested if PRMT5 and MSI2 inhibitors alone or in combination affected *c-MYC* mRNA in lymphoma cells. RNA-Seq analysis showed that treatment with GSK-591 and Ro alone did not affect *c-MYC* or *MSI2* at the mRNA level, however, the combination of these drugs subtly reduced *MSI2* mRNA (*p*-value < 0.02) but did not induce significant changes to *c-MYC* mRNA (Fig. 5A). Next, we tested how the inhibition of MSI2 and PRMT5 affects c-MYC protein abundance in B-cell lymphoma cells. In contrast to PRMT5 inhibitor at 12 and 24 h, where we observed an increase in c-MYC abundance, treatment with Ro resulted in reduced c-MYC. Combination of Ro and GSK-591 dramatically decreased c-MYC and MSI2 protein at both time-points (Fig. 5B). Moreover, MSI2 knockdown reduced c-MYC protein abundance that was further depleted upon dual treatment with GSK-591 (Fig. 5C). These findings confirm PRMT5 and MSI2 cooperate by maintaining c-MYC protein abundance.

To further study whether c-MYC plays a role in resistance to PRMT5 and MSI2 inhibition, we used the B-cell lymphoma cell line, P-4936 cells, that express c-MYC under the control of a tetracycline-inducible promoter. c-MYC depletion significantly increased the sensitivity to GSK-591 and Ro combination (Fig. 5D). Interestingly, c-MYC loss of expression resulted in the reduction of MSI2 abundance in P-4936 cells, suggesting that c-MYC and MSI2 form a feedback loop to regulate each other's expression (Fig. 5E). In contrast, overexpression of c-MYC significantly rescued cell viability of Z-138 cells treated with Ro or with the combination with GSK-591 (Fig. 5F).

We next examined if the reduction of c-MYC was due to direct MSI2 binding to *c-MYC* mRNA. We performed RNA immunoprecipitation (RNA-IP) with Z-138 cells treated with either vehicle, GSK-591, Ro, and the combination. We confirmed that MSI2 binds *c-MYC* mRNA in the control cells and GSK591-treated cells, but the treatment with Ro or the combination significantly impaired MSI2 binding to *c-MYC* RNA (Fig. 5G). Importantly, the total MSI2 abundance was reduced in the combination treated groups. To address whether PRMT5 and MSI2 regulates c-MYC translation, we performed luciferase assays using a reporter encoding the 3'UTR of *c-MYC*. Pharmacological inhibition or genetic deletion of either PRMT5 and MSI2 significantly decreased c-MYC luciferase activity (Fig. 5H, I, respectively), suggesting that direct binding of *MSI2* to the 3'UTR of *c-MYC* controls translation of c-MYC and that combination treatment further depletes c-MYC translation via the 3'UTR. These data suggest that MSI2 binding targets could be further altered by the combination treatment.

### HyperTRIBE to identify MSI2 targets mediating resistance to PRMT5 inhibition

To determine MSI2's direct targets and how Ro and GSK-591 affect MSI2 binding to these targets, we utilized MSI2-HyperTRIBE. We had recently found this analysis allows for the identification of the MSI2 binding network in hematopoietic stem cells and myeloid leukemia cells[31]. We overexpressed MSI2-ADAR, which is the human MSI2 fused to the catalytic domain of *Drosophila* ADAR (adenosine deaminase enzyme). In this fusion, ADAR marks MSI2 binding sites that can be identified by RNA-sequencing (Fig. 6A). Motif enrichment analysis revealed that the top 5 most significantly enriched MSI2 binding motifs in lymphoma cells match the previously identified MSI2 motifs[31,32] (Fig. 6B). Overexpression of MSI2-ADAR in Z-138 cells significantly increased (over 10-fold) the edit frequency on RNAs compared with the empty vector control (MIGR1) (*p*-value < 0.0001) (Fig. 6C). MSI2-HyperTRIBE identified 1991 target genes marked by 4692 significant edit sites in Z-138 cells. As expected, most of these edit sites (-92%) occurred in the 3'UTR region (Supplementary Fig. 5A).

To evaluate whether GSK-591, Ro, or the combination affects MSI2 binding to RNA, we overexpressed MSI2-ADAR and treated the cells with DMSO, Ro, GSK-591 or combination of Ro and GSK591. Consistent with Ro inhibiting MSI2 activity, we found a significant reduction in editing frequency as compared to vehicle controls (Fig. 6C). MSI2's direct targets were reduced with introduction of Ro (differential frequency ≤ −0.1; 3108 significant edit sites representing 1502 gene targets; *p*-value < 0.05), while PRMT5 inhibition did not yield significant changes to MSI2 edit sites. The combination of Ro and GSK-591 did not induce significant changes in MSI2 binding when compared with Ro alone (differential frequency ≤ −0.1; 3410 significant edit sites representing 1502 gene targets; *p*-value < 0.05), (Fig. 6D and Supplementary Data 9).

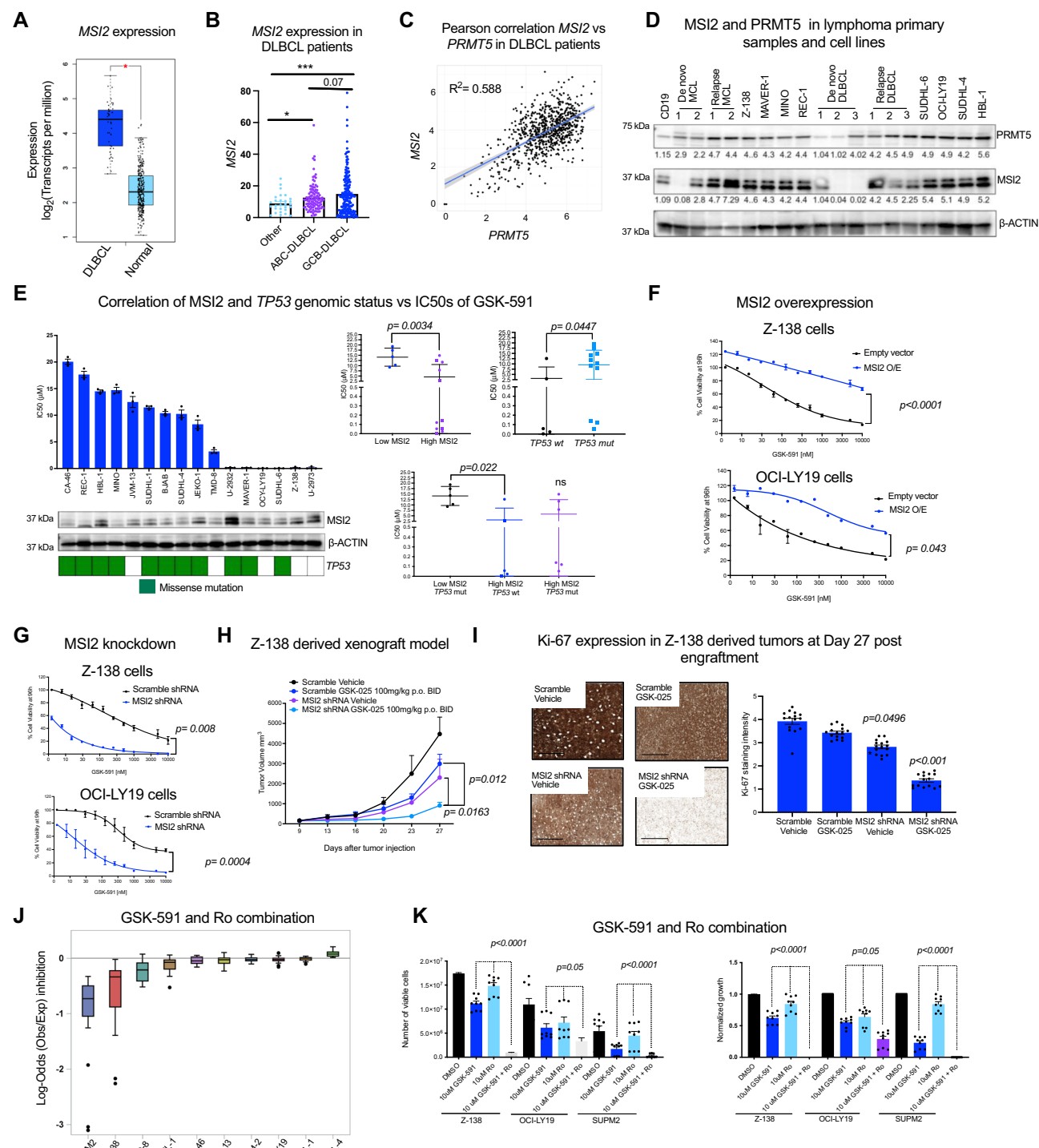

**Fig. 3 | MSI2 drives resistance to PRMT5 inhibition. A** *MSI2* expression (TPM) in primary samples from DLBCL patients ($n = 47$) (TCGA database in GEPIA) vs. normal B-cells ($n = 337$) (GTEX). (*$p$-value < 0.01, |Log2FC| Cutoff: 2). **B** *MSI2* levels (FKPM) in ABC, GCB-DLBCL and non-characterized subtypes from the BC Cancer Lymphoid Cancer Database ($n = 322$). **C** Pearson correlation coefficient of *PRMT5* and *MSI2* expression across 820 DLBCL patients. **D** Immunoblot of MCL and DLBCL de novo and relapsed primaries and cell lines vs. normal CD19+ B-cells from healthy donors. Similar results obtained in two different immunoblots. **E** IC50 values of GSK-591 at Day 8, MSI2 protein abundance (representative images of 3 different immunoblots) and *TP53* mutation status in 16 cell lines. Left. GSK-591 IC50s in low MSI2 abundance (below MSI2/ACTIN ratio) and high MSI2 (above MSI2/ACTIN ratio) cells. Right. IC50s of GSK-591 in *TP53* wt and mutant cells. Bottom. IC50s of GSK-591 in the indicated conditions. $p$-values calculated by one-tailed $t$ test. $n = 3$ analyzed in 3 independent experiments. **F** Cell viability of Z-138 and OCI-LY19 cells expressing FLAG-MSI2 or empty vector after 96 h GSK-591 treatment. Data represent mean ±

SEM, $n = 3$ replicates of 3 independent experiments. Two-sided ANOVA. **G** Z-138 and OCI-LY19 cells expressing control shRNA(shCtrl) or MSI2 shRNA. Error bars represent SEM of three different experiments. Two-sided ANOVA. **H** Z-138 scramble or MSI2 shRNA xenografts ($n = 5$/group) were treated with vehicle or GSK-025. Data are represented as mean ± SEM. $p$-values calculated by two-sided ANOVA. **I** Percentage of total sectional area positive for Ki-67 staining on the indicated conditions. Scale bars, 1000 μm. **J** Effect of the combination of GSK-591 and Ro on 10 lymphoma cell lines' viability. Maximum log-odds = 0.23 (additive) and minimum log-odds = −3.8 (synergy). Data represent mean ± SEM of three separate determinations. Cells were incubated with increasing concentrations of GSK-591 and venetoclax for 72 h. **K** Synergistic combination of GSK-591 and Ro in the indicated cells at 10 μM of each agent. Absolute viable cell numbers (left) and the normalized growth (right). Data represent mean ± SEM of three different experiments, $n = 3$ replicates. $p$-values were calculated by two-sided ANOVA.

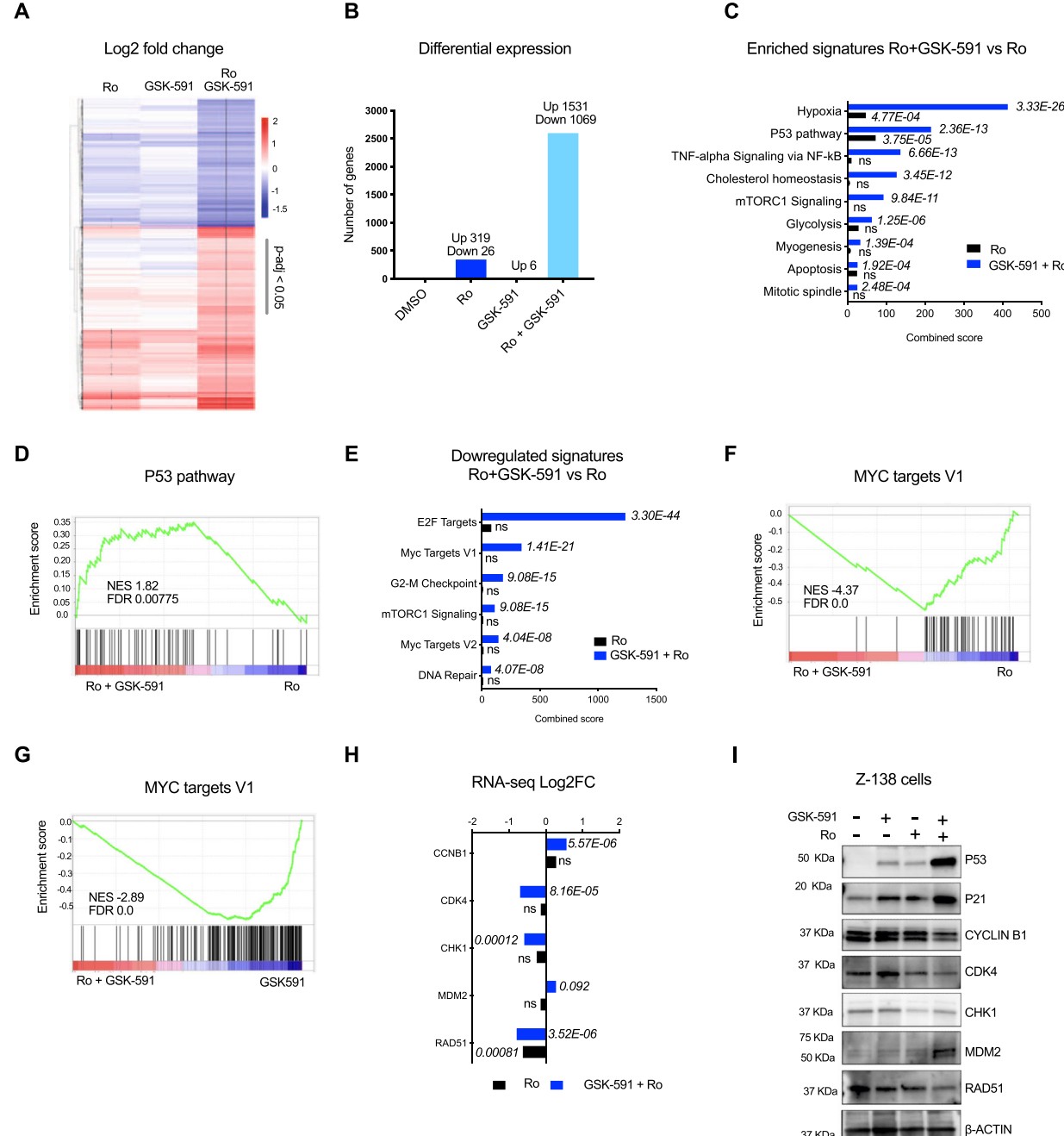

**Fig. 4 | Combination of Ro and GSK-591 induces synergistic transcriptome changes. A** Heatmap showing differential expression of the top 2600 significantly enriched (in red) and depleted (blue) transcripts in cells treated with 5uM of Ro, GSK-591 alone or in combination for 24 h. Values represent Log2FC, * indicate significant (adjust *p*-value < 0.05) DESeq genes, determined by one-sided Wilcoxon test. **B** Bar graph summarizing the number of significant DESeq genes from A in the indicated conditions. **C** Top significant enriched Molecular Signatures (MSigDB) in the upregulated transcripts (1531) upon drug combination vs. Ro alone using ENRICHR analysis. *p*-values were calculated by Fisher's exact test. **D** GSEA analysis of differentially expressed genes upon combination of GSK-591 and Ro. P53 signature was positively enriched in the combination vs Ro alone. **E** Top

significant enriched Molecular Signatures (MSigDB) in the downregulated transcripts (1069) upon drug combination vs. Ro alone using ENRICHR analysis. *p*-values were calculated by Fisher's exact test. **F**, **G** GSEA analysis of differentially expressed genes upon combination of GSK-591 and Ro. MYC targets V1 signature was negatively enriched in the combination vs Ro or GSK-591 alone, respectively. **H** mRNA expression of selected cell-cycle regulation genes upon Ro or combination treatment. *n* = 3 independent experiments; data represented as mean ± SEM, two-sided Student's *t* test. **I** Representative immunoblot analysis of the indicated cell-cycle regulator proteins. Z-138 cells were treated with 5uM of GSK-591 and Ro for 24 h. Beta-actin expression was used as a loading control. Similar abundance was observed in 2 different immunoblots. Source data are provided as a Source data file.

We then functionally annotated all the Ro-specific targets identified in the MSI2-HyperTRIBE and confirmed that most of the targets in B-cell lymphoma are enriched for pathways previously linked to MSI2 in other cellular contexts[18,33,34] (Fig. 6E and Supplementary Table T3). Additionally, gene ontology (GO molecular function) of Ro-targets identified pathways related to RNA metabolism and protein

degradation (Supplementary Fig. 5B and Supplementary Data 10). Overall, these data confirm Ro is blocking MSI2 binding to its targets in B-cell lymphoma cells, but GSK-591 alone or dual drug treatment did not affect MSI2 target profiles.

To determine whether the effects on gene expression were due to Ro-dependent MSI2 binding, we overlapped the differential

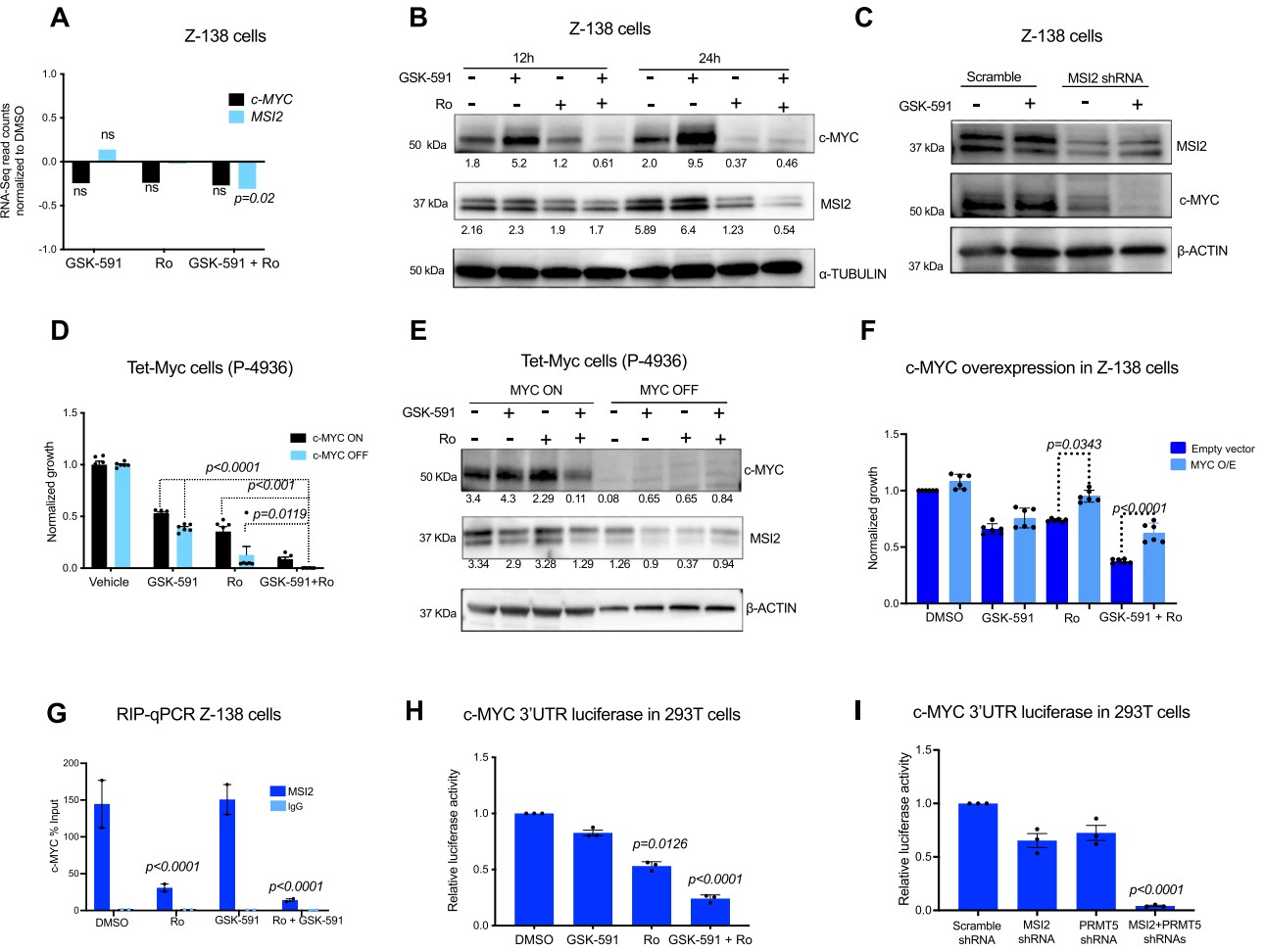

**Fig. 5 | MSI2 mediates PRMT5 resistance through control of c-MYC translation.**
**A** mRNA expression of *c-MYC* and *MSI2* from RNA-Seq experiment Fig. 4E, *n* = 3.
*p*-value 0.020 determined by two-sided Student's *t* test. **B** Representative immunoblots of c-MYC and MSI2 upon treatment with GSK-591 and Ro at the indicated times in Z-138 cells. Similar results were observed in <4 different experiments. **C** Representative immunoblots of c-MYC and MSI2 upon MSI2 knockdown. Similar results were observed in two independent experiments. **D** Effect of GSK-591 and Ro treatment on cell viability upon c-MYC depletion (c-MYC OFF) for 72 h in P-4936 cells. Data represent mean ± SEM of three different experiments, *n* = 3 replicates. **E** Representative immunoblots showing decreased MSI2 protein abundance, in P-4936 cells after 72 h of MYC depletion (MYC OFF). Similar results were observed in two independent experiments. **F** Effect of GSK-591 and Ro treatment on Z-138

cells overexpressing empty vector or c-MYC after 96 h. Values are the mean ± SEM of three separate experiments. **G** qRT-PCR of recovered RNA from MSI2 RNA-IP from Z-138 cells treated with 5 µM GSK-591 and/or Ro for 24 h. c-MYC mRNA enrichment is shown as the percentage (IP/input) and normalized to DMSO ± SEM of three independent experiments. IgG served as a non-specific binding control. *p*-value determined by two-sided Student's *t* test. **H** Luciferase reporter assay using MYC 3′UTR in 293T cells. 293T cells were treated with GSK-591 and/or Ro for 24 h. Luciferase activity was normalized using firefly values. *n* = 3 independent experiments. **I** Luciferase reporter assay using MYC 3′UTR in 293T cells. 293T cells were transfected with control or MSI2 and PRMT5 shRNA constructs. Luciferase activity was measured 48 h post-transfection and was normalized using firefly values. *n* = 3 independent experiments.

expression dataset (Fig. 4) with the Ro-specific targets from MSI2-HyperTRIBE dataset. We found that the overexpression of MSI2-ADAR does not induce significant changes in the transcriptome in comparison to the empty vector (MIGR1) (Supplementary Fig. 5C). However, Ro-dependent MSI2 binding targets were found to be significantly but modestly reduced after Ro or GSK-591 treatment when compared to the control (Fig. 6F and Supplementary Fig. 5D).

Although MYC is a well-characterized target of MSI2 and PRMT5, it was not among the top hits either in the MSI2 HyperTRIBE or the CRISPR-Cas9 screen. As we previously hypothesized, the rapid turnover of *c-MYC* could prevent its detection in the HyperTRIBE assay[31]. Nevertheless, the HyperTRIBE in B-cells lymphoma identified MSI2 canonical targets *MYB*, *IKZF2*, *NUMB*, and *SMAD9* (Supplementary Data 9). As previously reported, MSI2 inhibition induced modest or no effect on *MYB* mRNA or *IKZF2* mRNA, the protein abundance of these Ro-targets was markedly reduced upon Ro and combination treatment (Supplementary Fig. 5E, F). These data support that MSI2 binding

impacts mRNA metabolism and may also result in additional effects on the translation of these targets.

## BCL-2 is a target of PRMT5/MSI2 axis and contributes to resistance to PRMT5 inhibition

To identify additional MSI2 direct targets that promote resistance to PRMT5 inhibition in B-cell lymphoma, we overlapped: 1) differentially expressed genes in Z-138 cells treated with Ro, GSK-591 and the combination (DeSeq data), 2) the Ro-specific MSI2 targets (HyperTRIBE) and 3) genes that confer resistance to PRMT5 inhibition (genome-wide CRISPR screen). The analysis revealedt the anti-apoptotic protein BCL-2, hexamethylene-bis-acetamide (HMBA)-inducible protein 1 (HEXIM1) and the spindle and kinetochore-associated protein 2 (SKA2) as candidates that could be part of the PRMT5-MSI2 axis (Fig. 7A and Supplementary Data 11).

To validate these additional targets, we performed RNA-IP and confirmed that MSI2 binds *BCL-2* mRNA and treatment with Ro or with the combination significantly blocked MSI2 binding to *BCL-2* (Fig. 7B).

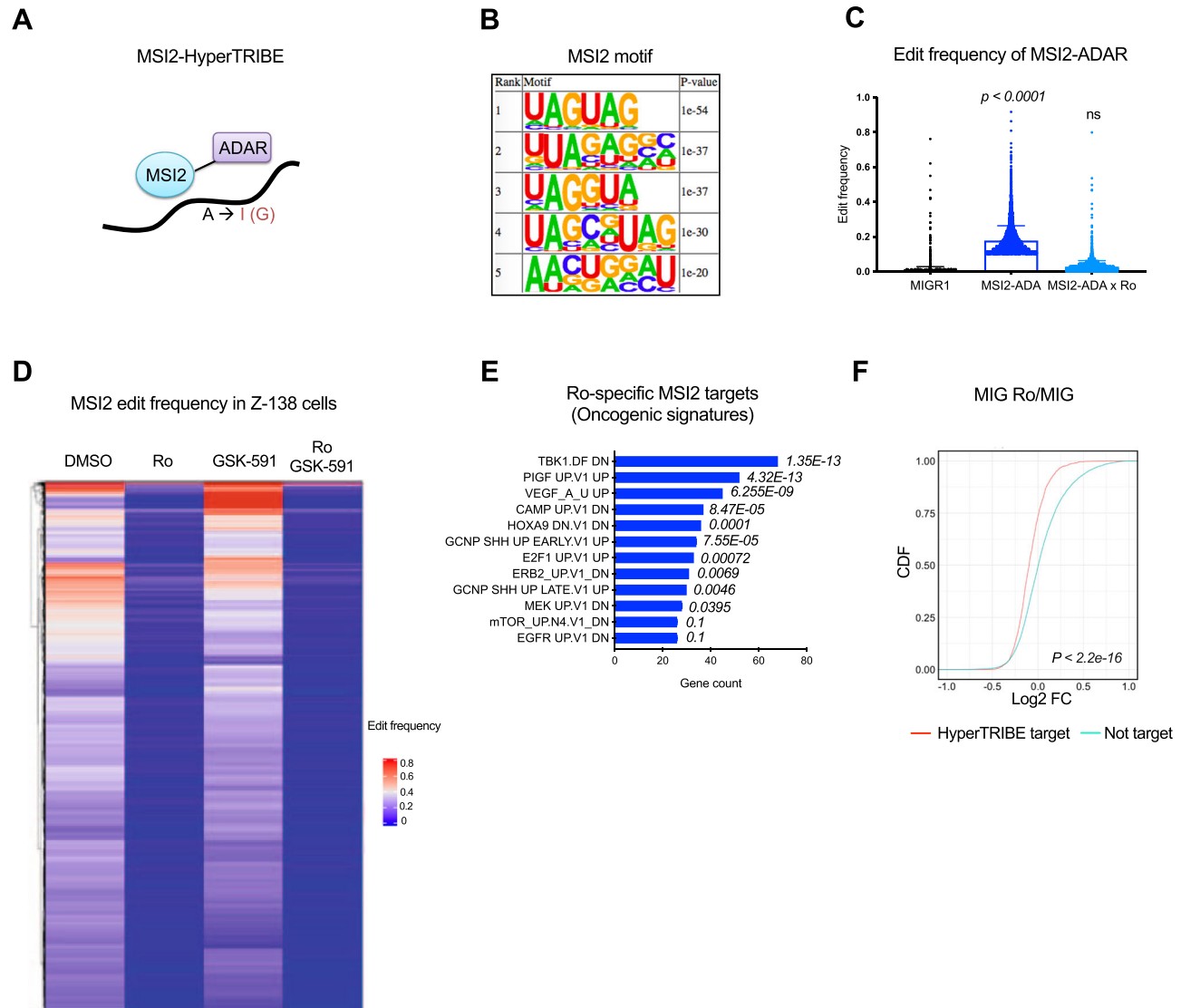

**Fig. 6 | Ro blocks MSI2 mRNA binding and MSI2 binding specificity is PRMT5 independent.** **A** Schematic illustration showing the MSI2 fusion to the ADAR catalytic domain. **B** HOMER motif enrichment was performed on ±100bp window flanking both sides of the edit site, totaling a 200 bp window per edit site. Top 5 motifs and the corresponding *p*-values are shown, statistical analysis by Fisher's test. **C** Edit frequency on mRNAs in Z-138 cells overexpressing empty vector (MIG) or MSI2-ADAR and MSI2-ADAR treated with Ro for 24 h. Unpaired Kruskal−Wallis test; *p*-value < 0.0001. **D** Heatmap of MSI2 edit frequency in Z-138 cells treated with Ro, GSK-591, or the combination for 24 h. HyperTRIBE data was filtered to include

MSI2-ADAR significant edit sites (adjusted *p*-value < 0.05, diff.frequency < = −0.1, fpkm > = 5) and MSI2-ADAR + Ro significant edit sites (adjusted *p*-value < 0.05, diff.frequency > = 0.1, fpkm > = 5). Only genes more significantly edited (beta-binomial test) are plotted. **E** Top significant enriched Oncogenic signatures in Ro-specific targets (1502 genes) using ENRICHR analysis Fisher exact test. **F** CDF plots showing the distribution of mRNA abundance changes in MSI2 HyperTRIBE targets (filtered by *p*-value < 0.05, diff.frequency < = −0.1, and fpkm > = 5) upon Ro treatment. A two-sided KS test was used to calculate *p*-value. Source data are provided as a Source data file.

Furthermore, MSI2 binds *HEXIM1* and *SKA2* mRNA, however, Ro alone and the combination perturbed only MSI2 binding to *SKA2* (*p*-value < 0.0001) and did not affect the binding to *HEXIM1* (Supplementary Fig. 6A). Similar to what we observed with c-MYC, the reduced binding is likely from decreased MSI2 abundance in the combination conditions. RNA-Seq analysis found that the combination of GSK-591 and Ro subtly downregulated *BCL-2* (*p*-value < 0.005) and *HEXIM1* (*p-value* < 0.0001) mRNAs, while *SKA2* was modestly upregulated upon combination treatment (*p*-value < 0.0004) (Supplementary Fig. 6B). Moreover, immunoblotting analysis demonstrated that combination of Ro and GSK-591 dramatically reduced BCL-2 and SKA2 protein abundance, while there was no significant change in HEXIM1 (Fig. 7C and Supplementary Fig. 6C). The genetic depletion of MSI2 and/or PRMT5 using hairpin RNAs (shRNAs) induced similar effects as the pharmacological inhibition on the target's expression, c-MYC, SKA2,

and BCL-2, while HEXIM1 protein abundance was not affected (Supplementary Fig. 6D). Importantly, we found that MSI2 depletion and PRMT5 inhibitor synergistically decreased c-MYC and BCL-2 protein abundance in vivo (Fig. 7D and Supplementary Fig. 6E). These data suggest that MSI2 binds and regulates *c-MYC*, *BCL-2*, and *SKA2*.

*BCL-2* is highly expressed in B-cell lymphomas, and a great majority of these cells are dependent on BCL-2 for survival[35]. Thus, we focused on the study of the functional relevance of BCL-2 as a downstream target of the PRMT5/MSI2 axis. First, we wanted to assess the functional requirement of BCL-2 in the activity of Ro and GSK-591. To this end, we generated isogenic BCL-2 stable knockout cells by transducing Z-138 and OCI-LY19 Cas9 cells with an sgRNA targeting BCL-2. Z-138 and OCI-LY19 BCL-2 KO cells were significantly more sensitive to GSK-591 alone, Ro alone, and in combination compared with the BCL-2 expressing cells (Fig. 7E and Supplementary Fig. 6F). To further

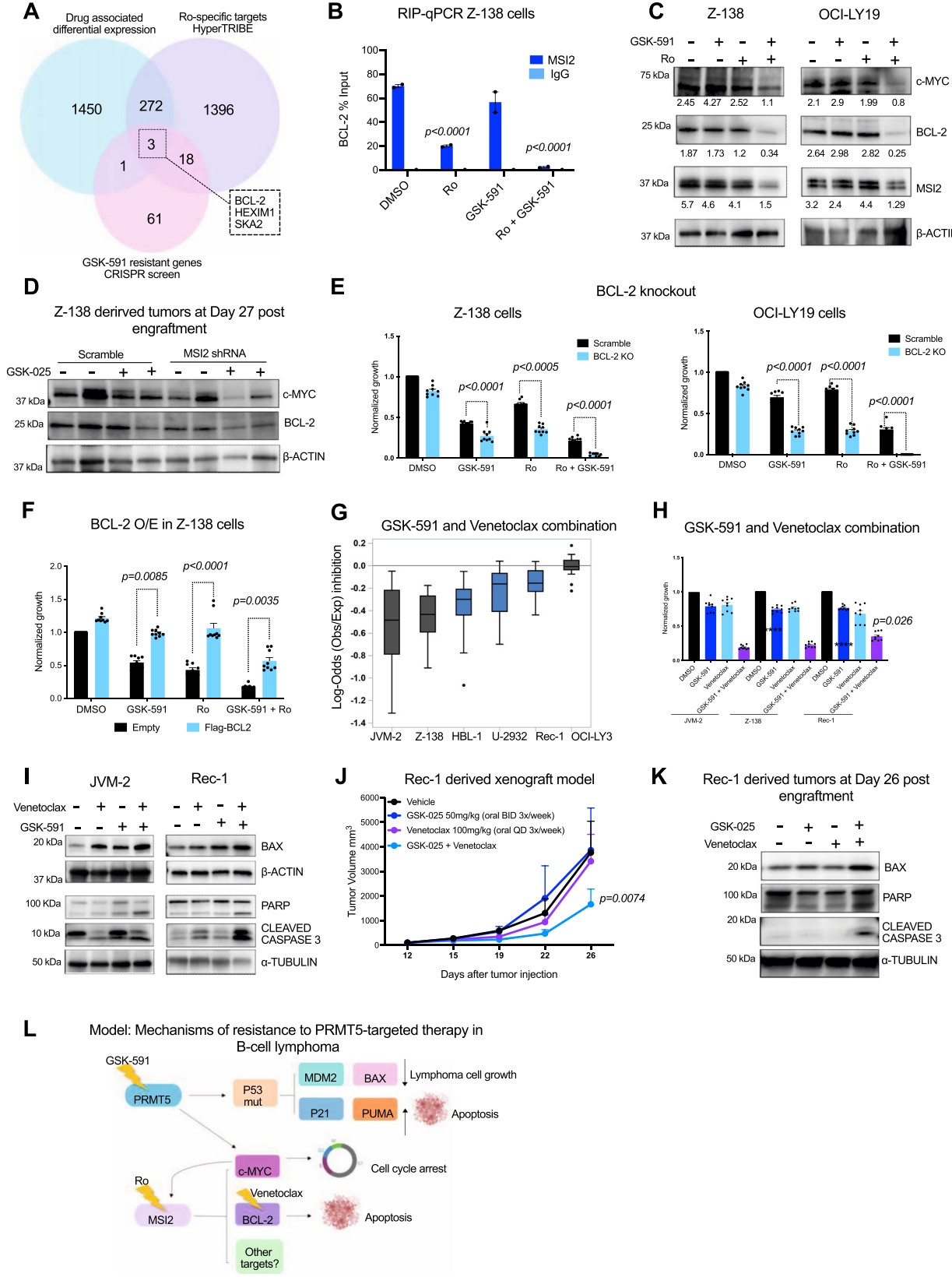

support this axis, we found that forced expression of BCL-2 conferred resistance to PRMT5 inhibition in Z-138 cells (Fig. 7F and Supplementary Fig. 6G).

To study whether BCL-2 inhibition is therapeutically relevant in the context of PRMT5-targeted therapy in lymphoma, we used venetoclax, a potent and selective BCL-2 inhibitor, which is FDA-approved for the treatment of patients with chronic lymphocytic leukemia and small lymphocytic leukemia[36]. In vitro viability assays show venetoclax enhances the activity of GSK-591 in most cell lines tested, even in the most GSK591-resistant cell lines, DB and NUDHL-1, which carry the $TP53^{R248W}$ mutation (Fig. 7G, H, Supplementary Fig. 6H and Source Data). Furthermore, the combination therapy was the most potent,

**Fig. 7 | MSI2 regulates BCL-2 translation and GSK-591 synergizes with veneto-clax inducing apoptosis in vitro and in vivo. A** Venn diagram showing overlapped genes, 1 by RNA-Seq; 2: HyperTRIBE targets; 3: resistant genes to GSK-591 (genome-wide CRISPR screen). **B** *BCL-2* qRT-PCR from MSI2 RNA-IP is shown as the percentage (IP/input) and normalized to DMSO ± SEM of two independent experiments. IgG served as a non-specific binding control. **C** Representative immunoblots of Z-138 cells treated with GSK-591 and Ro for 72 h in. Similar results were observed in 3 independent experiments. **D** Representative immunoblot showing c-MYC and BCL-2 expression in MSI2 shRNA mice treated with GSK-025. Similar abundance was observed in 2 independent immunoblots. **E** Effect of GSK-591 and Ro treatment on BCL-2 KO cells in Z-138 and OCI-LY19 cells. Data represent mean ± SEM of three different experiments (*n* = 3). *p*-values calculated by 2way ANOVA **F** Effect of GSK-591 and Ro treatment on Z-138 cells overexpressing empty vector or FLAG-BCL-2 after 72 h. Data represent mean ± SEM of three different experiments, *n* = 3. *p*-values calculated by 2way ANOVA. **G** Boxplot graph showing the effect of the combination of GSK-591 and venetoclax on 3 *TP53* wt (gray) and 3 *TP53* mutant (blue) cells. Maximum Log-odds = 0.2 (additive); minimum log-odds = −1.39 (synergy). Data represent mean ± SEM of three separate determinations. **H** Bar graph showing the synergistic combination of GSK-591 and venetoclax in the indicated cells at 10uM of each agent. Data represent mean ± SEM of three different experiments (*n* = 3). ****p* < 0.0001 calculated by 2way ANOVA. **I** Representative immunoblot showing the effect of the drug combination in JVM-2 (*TP53* wt) and Rec-1 (*TP53* mutant) after 72 h of treatment. Similar protein expression levels were observed in 2 independent immunoblots. **J** Rec-1 xenografts (*n* = 5/group) were treated with vehicle, GSK-025, venetoclax or the combination. *p*-value was calculated by 2way ANOVA. **K** Representative immunoblot showing the effect of GSK-025 and venetoclax combination on the indicated proteins in vivo. Similar abundance was observed in 2 independent immunoblots. **L** Model of mechanisms of resistance to PRMT5 inhibition in B-cell lymphoma.

inducing apoptosis (Supplementary Fig. 6I) as confirmed by the cleavage of caspase 3 and PARP, when compared to each compound alone (Fig. 7I).

To evaluate the efficacy of the combination of GSK-591 and venetoclax in vivo, we generated a human MCL xenograft model using the Rec-1 cell line, that is resistant to both single agents. Monotherapy with GSK-591 or venetoclax did not affect tumor progression even after 3 weeks of treatment. However, the co-administration of GSK-591 and venetoclax reversed the resistance to the single agents, significantly reducing the tumor volume without adding toxicity (Fig. 7J and Supplementary Fig. 6J). Similar to the in vitro results, the combination of GSK-591 and venetoclax resulted in caspase 3 and PARP cleavage and upregulation of BAX in vivo (Fig. 7K).

Based on our results, we propose two mechanisms of resistance to PRMT5 inhibition, the first mediated by loss of function of *TP53*, which can be caused by its genetic deletion or the hotspot mutation *TP53^R248W^*. *TP53* LOF impairs the upregulation of its targets, P21, BAX, PUMA, and consequently, the apoptosis mediated by PRMT5 inhibition. The second mechanism of resistance is driven by MSI2. In the most aggressive forms of lymphoma, MSI2 is overexpressed and cooperates with PRMT5 in maintaining c-MYC and BCL-2, which confers resistance to PRMT5 inhibition. Therefore, to inhibit the PRMT5/MSI2/c-MYC/BCL-2 axis we proposed two drug combination strategies; the first one consists of targeting PRMT5 in combination with a MSI2 inhibitor, which results in loss of c-MYC and BCL-2 translation cell-cycle regulators in lymphoma cells. The second strategy is the combination of GSK-591 with the BCL-2 inhibitor, venetoclax, which induces apoptosis of lymphoma cells (Fig. 7L).

## Discussion

Our study uncovers mechanisms of resistance to the selective PRMT5 inhibitor, GSK-591, and proposes drug combination strategies that could better leverage use of PRMT5 inhibitors in clinical settings. Depletion experiments confirm that PRMT5 protein expression is essential for B-cell lymphoma survival. Importantly, the catalytic activity was essential for survival since the PRMT5 catalytic dead mutant could not rescue this function. However, lymphoma cells had variable sensitivities (45% of lymphoma cell lines tested were less sensitive) to pharmacological inhibition of the methyltransferase. Additionally, PRMT5 inhibition in vivo exhibited modest anti-tumor activity. To help understand which pathways or genes contribute resistance and sensitivity to PRMT5 inhibition, we performed a genome-wide CRISPR/Cas9 screen.

Previously, Gao et al., performed a CRISPR/Cas9 screen in the presence of a PRMT5 inhibitor using a small library targeting epigenetic regulators (1016 in total) in non-small cell lung cancer cells. They identified MEP50 and PRMT1 as key molecular targets of GSK-591[37]. MEP50 is a known cofactor of PRMT5, and enhances PRMT5 binding affinity to its substrates[38], hence it is not surprising that MEP50 was

identified as a top hit in this screen. Likewise, PRMT1 shares common substrates with PRMT5, thus the dual inhibition of PRMT1 and PRMT5 results in potent anti-tumor activity in different cancer models[39,40]. However, our unbiased screen did not identify MEP50 nor PRMT1 as top targets, suggesting the existence of cell context dependencies for GSK-591 action.

Our screens identified *TP53* as one of the most sensitizing genes to GSK-591 in lymphomas. PRMT5 regulates P53 through maintaining the splicing fidelity of MDM4 and thus negatively regulates P53 function[41,42]. Additionally, PRMT5 may directly methylate P53 on R333, R335 and R337, preventing its oligomerization with MDM2, which prevents *TP53* binding to its target genes[42]. Our data support a link between PRMT5 and P53 with several experiments. First, we observed that most cell lines with wild-type *TP53* are sensitive to GSK-591. Second, we demonstrated that P53 is required for GSK-591 activity because its loss conferred resistance to PRMT5 inhibition. Finally, reintroduction of wild-type *TP53* in cells harboring *TP53* missense mutations was sufficient to restore lymphoma cells' sensitivity to GSK-591. These data indicate that GSK-591 inhibits PRMT5 and the functional consequences are mediated through P53.

*TP53* genetic alterations are found throughout the gene structure, but the functional relevance for each mutation has not been fully characterized and could be dependent on cell context. Some mutations are suggested to have either a loss-or gain-of-function[43-45]. However, mutations localized in the DNA-binding domain are likely to act like loss-of-function mutation as they abrogate *TP53* transcriptional activity[46,47]. We found the R248W mutation of *TP53* is associated with resistance to GSK-591 in vitro and in vivo. The hotspot *TP53^R248W^* mutation induces dominant negative activity preventing TP53 binding to its target promoters and correlates with shorter survival in patients across different types of cancer[48-50]. GSK-591 failed to induce *TP53* targets, PUMA, P21 and BAX in the *TP53^R248W^* isogenic cell lines, even though P53 was upregulated. In agreement with our results, the chemotherapeutic agent, daunorubicin, induces P53 protein expression in *TP53^R248Q^* cells, however, this protein is not able to recruit targets, such as P21[48]. Thus, we hypothesize that PRMT5 might still be able to bind and methylate *TP53^R248W^*, therefore, GSK-591 treatment results in the upregulation of a dominant negative form of TP53 that mimics its deletion. Our findings have clinical implications as *TP53^R248W/Q^* is the most common site-specific mutation in all cancers, thus it could be used for patient selection in the ongoing and future clinical trials with PRMT5 inhibitors.

Our CRISPR/Cas9 screens identified the RNA-binding protein MSI2 as the top driver of resistance to PRMT5 inhibition. We found that *MSI2* is highly expressed in DLBCL patients and strongly correlates with *PRMT5* expression. Elevated *MSI2* expression is considered a biomarker of poor outcome in multiple hematological diseases, such as chronic myeloid leukemia (CML), acute myeloid leukemia (AML), myelodysplastic syndrome (MDS), and chronic lymphocytic leukemia

(CLL)[14,15,19,51-53]. We observed that MSI2 depletion reduced about 50% tumor growth and resistant tumor cells escaped due to inefficient MSI2 knockdown in a lymphoma xenograft model. Considering this, it would be interesting to conduct studies on B-cell lymphoma mouse models to determine whether MSI2 is required for both lymphoma initiation and progression.

Previously, it has been shown that inhibition of MSI2 with Ro reduced disease burden in AML and CLL models[19,26]. We demonstrate that the combination of Ro with GSK-591 induces synergistic anti-proliferative activity and proves to be a viable therapeutic strategy to target B-cell lymphomas. Previously, we found Ro reduced binding of MSI2 to selected targets using RNA-IP (*TGFBR1*, *MYC*, *SMAD3*, and *P21*)[26]. Our HyperTRIBE analysis found that Ro inhibited MSI2 binding to its targets regardless of the frequency of binding. It is also not clear why some targets were inhibited more than others. Over 1500 mRNAs were identified as Ro-specific targets in B-cell lymphoma. These Ro-targets were enriched for pathways previously linked to MSI2 in hematopoiesis and leukemogenesis. Thus, our data supports the need to develop more potent and clinically relevant molecules targeting MSI2 in multiple hematological malignancies.

MSI2 drives leukemia progression and stem cell renewal through a variety of mechanisms involving multiple pathways. We previously found that MSI2 maintains *MLL*-leukemia self-renewal programs by retaining the translation of *HOXA9*, *c-MYC*, *MYB*, and *IKZF2*[18,31]. MSI2 directly binds and maintains *TETRASPANIN 3* (*TSPAN3*), branched-chain amino acid aminotransferase 1 (*BCAT1*) and FMS-like tyrosine kinase 3 (*FLT3*)[54,55]. Besides promoting translation, MSI2 mediates repression of its targets, including *NUMB* and *P21*[15,26].

Using HyperTRIBE, we identify direct targets of MSI2 that cooperate with the resistance mechanism to GSK-591 in B-cell lymphoma. Consistent with previous studies, in B-cell lymphoma, MSI2 binds to canonical targets such as *MYB* and *IKZF2*, and Ro treatment resulted in their loss of protein abundance.

PRMT5-mediated arginine methylation of RBPs is required for optimal binding activity[56,57]. PRMT5 depletion and genetic deletion block the binding of splicing factors, such as SRSF1 and SRSF2, to their mRNA targets[58,59]. Moreover, another study demonstrated that the combination of PRMT5 and PRMT1 inhibitors is synergistic in AML due to the inhibition of the arginine methylation of a large compendium of RBPs[40]. However, MSI2 was not identified among PRMT5's methylation targets in these studies, which is consistent with the fact that PRMT5 inhibition alone did not affect MSI2 RNA-binding activity in our study. Nevertheless, regulation of multiple RBPs could lead to a combinatorial effect when both MSI2 and these targets are affected.

The differential expression analysis revealed the combination of GSK-591 and Ro is synergistic, inducing global changes in the transcriptome of B-cell lymphoma cell lines. Thus, the combination resulted in greater downregulation of signatures associated with cell cycle, c-MYC targets and upregulation of the P53 pathway. Previous studies have demonstrated that both PRMT5 and MSI2 maintain c-MYC expression in cancer cells through independent mechanisms, but we found MSI2 and PRMT5 cooperate by retaining c-MYC translation. Both pharmacological inhibition and genetic depletion of PRMT5 and MSI2 resulted in downregulation of BCL-2. Additionally, we found c-MYC and BCL-2 are critical components of the PRMT5/MSI2 axis. It would be interesting to investigate whether the mechanism of resistance to PRMT5-targeted therapies mediated by the oncogenic axis, MSI2/c-MYC/BCL-2, could be extrapolated to the other cancers. These results suggest that PRMT5 and MSI2 cooperate in sustaining the shared translation of common oncogenic signaling pathways, which underscores the relevance of the dual targeting to prevent feedback mechanisms of resistance induced by the single agent.

Venetoclax is the only selective BCL-2 inhibitor approved for treatment of patients with chronic lymphocytic leukemia and small lymphocytic leukemia. However, we and others have described multiple mechanisms of resistance to BCL-2 inhibition in different types of cancer, including B-cell lymphomas[60-62]. Combination of GSK-591 and venetoclax resulted in potent anti-proliferative activity across different lymphoma subtypes and even in the most resistant lymphoma cells, lower doses of both drugs were required in combination, which could help to prevent potential off-target toxicity.

In summary, our study provides data for the role of the PRMT5/MSI2 axis in regulating the key lymphoma drivers, c-MYC and BCL-2. Of major clinical relevance, we demonstrated that *TP53* LOF and MSI2 expression may be useful for patient stratification in clinical studies with PRMT5 inhibitors. Furthermore, we provide a strong mechanism-based rationale for the therapeutic use of PRMT5 inhibitors in combination with MSI2 or BCL-2 inhibitors.

## Methods

### Cell lines, primary samples, and reagents

Primary patient samples were collected and approved by the Institutional Review Boards with the Biospecimen collection/banking study 06-107 and 12-245. The use of the samples for research purposes is covered under the Biospecimen research protocol 16–266.

Human MCL-derived cell lines Z-138, Rec-1, Jeko-1, Mino, JVM-2, JVM-13, and BL-derived cell lines Raji, EB-1, Daudi, Ramos and CA-46 were obtained from ATCC (American Type Culture Collection). Human DLBCL-derived cell lines SUDHL-4, SUDHL-6, SUDHL-8, SUDHL-10 OCI-LY19, DB, NUDHL-1, U-2973, OCI-LY3, U-2932, Ri-1 and OCI-LY10, Hodgkin's cell lines L-428, HDLM-2 and the MCL cell line, Maver-1, were obtained from the DSMZ-German Collection of Microorganisms and Cell Cultures, Department of Human and Animal Cell Cultures (Braunschweig, Germany). Cell lines HBL-1, TMD-8, SUP-M2, SUDHL-1, Karpas-299 and BJAB are not commercially available, and were kindly provided by Dr. R.E. Davis (MD Anderson Cancer Center, Houston, TX). P-4936 cells (Pajic et al., 2000; Zeller et al., 2006) were provided by Dr. J. Zhang (Thompson lab, Memorial Sloan Kettering Cancer Center, New York, NY). Cell lines were fingerprinted by STR analysis at the Integrated Genomic Operation Core, IGO (Memorial Sloan Kettering Cancer Center, New York, NY). All cell lines, PDX samples and primary samples were authenticated using a targeted deep sequencing assay of 585 cancer genes (HemePACT) at the IGO (MSKCC). Cell lines were cultured in RPMI 1640 medium supplemented with 10%–20% heat-inactivated fetal bovine serum (Hyclone, GE Healthcare Life Sciences [cat. SH30396.03]), 1% L-glutamine, and penicillin-streptomycin in a humid environment of 5% $CO_2$ at 37 °C.

$CD19^+$ Naive B cells were obtained from healthy donors. PBMC were isolated by Ficoll (GE Healthcare) density centrifugation and labeled with CD19 Microbeads (MACS Cat. 130-050-301).

Lymphoma patient cells were cultured in RPMI 1640 medium supplemented with 10% heat-inactivated fetal bovine serum (Hyclone, GE Healthcare Life Sciences [cat. SH30396.03]), 1% L-glutamine, and penicillin-streptomycin in a humid environment of 5% $CO_2$ at 37 °C.

GSK3235025 (GSK-025) used for the in vivo studies and GSK3203591 (GSK-591) for the in vitro experiments were provided by GlaxoSmithKline through MTA. Venetoclax was purchased from Selleckchem (Houston, TX). Ro 08-2750 was purchased from Tocris.

### CRISPR-Cas9 knockout screens and data analysis

For the genome-wide CRISPR screen, lentiviral particles were made from the Brunello library (Addgene 73179) that contains 76,441 sgRNAs, targeting each gene with 4 different guides, and 1000 non-targeting control sgRNAs.

For the validation CRISPR-Cas9 screen, a CRISPR KO library was designed containing 2020 sgRNAs targeting the top-ranked 316 enriched hits and 89 depleted hits from the genome-wide CRISPR screen, for a total of 405 genes (using the 4 sgRNAs per gene from the Brunello library), plus 200 non-targeting and 200 safe-harbor sgRNAs.

Z-138 (for the genome-wide CRISPR screen) or OCI-LY19 (for the validation CRISPR screen) cells expressing Cas9 were infected with the library for 48 h at a multiplicity of infection (MOI) of 0.3. Two infection replicates were performed. Two days after the infection, a total of 100 $\times 10^6$ cells (estimated library coverage of about 1000 cells per guide) were selected for 3 days with 4 µg/mL puromycin to enrich for cells carrying sgRNAs. Post-selection, 80 $\times 10^6$ cells were sequenced to identify genes critical for survival, T0 control. Remaining cells were expanded for 3 days. Then, cells were treated with DMSO or GSK-591 (IC80) for 8 days and survived cells were harvested.

Genomic DNA was isolated from all samples, and the sgRNA sequences were amplified by PCR (Supplementary Table T4) and sequenced on a HiSeq 2500 (Illumina). Copy numbers of sgRNAs were normalized in edgeR by Trimmed Mean of M-values (TMM). Then, the Correlation Adjusted Mean RAnk (CAMERA) function was used to calculate the statistical parameters (log2FC, p-values and FDR) at the gene level. This function implements a correlation-adjusted version of the Wilcoxon–Mann–Whitney test[63]. The sensitizing and resistant hits rank was obtained by applying two criteria: differential expression (adjust p-value < 0.1 and $log_2FoldChange < −2$ for depleted genes and $log_2FoldChange > 2$ for enriched genes) and the abundance of sgRNAs in the same direction depletion or enrichment (number of sgRNAs per gene > = 3).

## CRISPR knockout cell lines generation

Z-138 and OCI-LY19 cells were transduced with lentiCas9-Blast from Addgene (Cambridge, MA). 48 h after transduction cells were selected using 10 and 12 µg/ml blasticidin, respectively, for 5 days. Puromycin resistance cassette from LentiGuide-Puro vector backbone (Addgene 52963) was swapped for GFP and the following guide RNA sequences: TP53 KO1: 5′-GATCCACTCA CAGTTTCCAT-3′; TP53 KO2: 5′-GAGCGCTGCTCAGATAGCGA-3′; BCL-2 KO: 5′-GGCCTTCTTTGAGTTCGGTG-3′ were cloned into it. Z-138 and OCI-LY19 cells were transduced with lentiviral sgRNAs particles and 3 days post-transduction GFP-positive cells were sorted. Successful knockout was confirmed by immunoblotting.

## TP53$^{R248W}$ (c.741_742CC > TT) mutation knock-in

gRNAs targeting *TP53* R248 genomic region (hg38 chr17:7674171-7674270) were designed by GuideScan. gRNA sequences most proximal to R248 were cloned into lenti-guide-dsRed vector backbone (Supplementary Table T4). 4 µg gRNA_2 vector and 8 µg of repair template ssDNA (Supplementary Table T4) were electroporated into 800,000 Z-138 Cas9 using SG Cell Line 4D-Nucleofector X Kit (Lonza) according to manufacturer recommendations (4D-Nucleofector X Unit, Pulse Code CA-137). 48 h after electroporation, cells were enriched for dsRed by FACS as well as single-cell sorted into 96-well plates for clonal line generation. Genomic DNA (gDNA) was extracted from dsRed sorted cell pools. Primer pair p53ex7F1 & p53intR1 (Supplementary Table T4) were used for PCR with gDNA as template input for amplicon sequencing (GENEWIZ).

Sanger Sequencing of PCR products from 26 clonal lines were analyzed for HDR. gDNA from cells grown in 96-well plates were extracted. Primer pair p53ex6F2 and p53intR2 were used for PCR of HDR target region and for Sanger sequencing we used M13R2 primer (Supplementary Table T4).

Sequences of target region from potential *TP53*$^{R248W}$ knock-in clones were further validated by TOPO TA cloning (TOPO TA Cloning Kit, Thermofisher) and Sanger sequencing at the IGO (MSKCC).

## Immunoblotting

Immunoblot analyses were performed following standard procedures as described[60]. The following antibodies for western blotting were purchased from Abcam: PRMT5 (cat. ab31751, dilution 1:1000), MSI2 (cat. ab76148, dilution 1:2000), H3R2me2 (cat. ab194684, dilution 1:1000), H4R3me2 (cat. ab5823, dilution 1:500), HEXIM1 (cat. ab25388, dilution 1:1000) and NOXA (cat. ab13654, dilution 1:500) were purchased from Abcam. P53 (cat. 2527, dilution 1:1000), c-MYC (cat. 9402, dilution 1:1000), BCL-2 (cat. 4223, dilution 1:5000), BAX (cat. 2774, dilution 1:1000), MEP50 (cat. 2823, dilution 1:1000), SmD3me2 (cat. 13222, dilution 1:1000), PARP (cat. 9542, dilution 1:1000), Cleaved caspase-3 (cat. 9661, dilution 1:1000), P21 (cat. 2947, dilution 1:5000), PUMA (cat. 12450, dilution 1:1000), MDM2 (cat. 86934, dilution 1:1000) CYCLIN B1 (cat. 12231, dilution 1:1000), CDK4 (cat. 12790, dilution 1:1000), CHK1 (cat. 2360, dilution 1:1000), RAD51 (cat. 8875, dilution 1:500), GFP (cat. 2956, dilution 1:1000) and α-TUBULIN (cat. 3873, dilution 1:10000) were purchased from Cell Signaling Technology. β-ACTIN (cat. 5316, dilution 1:10000) and anti-FLAG (cat. F3165, dilution 1:5000) were purchased from Sigma-Aldrich. SKA2 (cat. PA5-20818, dilution 1:500) was purchased from Invitrogen.

## In vitro proliferation assay and combination assay

Cells were seeded in 24-well plates at 2.5 $\times 10^5$ cells/well with either vehicle (DMSO 0.1%) or increasing concentrations of drugs. Viable cell number was determined at 4- and 8-days drug incubation. On days of cell counts, the media and GSK-591 were replaced and cells split back to a density of 2.5 $\times 10^5$ cells/well. Cell viability was assessed using CellTiter-Glo Luminescent assay (Promega). 80 µL of the cultured cells were transferred to opaque, white bottom 96-well plates and mixed with 20 µL of CellTiter-Glo reagent. The mixture was incubated for 30 min at RT and read using EnSight Multimode plate reader (PerkinElmer) for luminescence. Data was normalized as percent viability and graphed by non-linear regression curves in Graph Pad PRISM 7.0.

To establish the interaction between two drugs in combination assays, we calculated the log-odds; the value: *log (observed growth inhibition / product of the 2 fractional growth inhibitions)*. A log-odds value of zero indicates that the combination treatments are additive; a negative value indicates synergy between the drugs and a positive value indicates antagonism.

## DNA constructs

pBabe-puro-Flag enconding PRMT5$^{G367A/R368A}$ and PRMT5 wild type were a gift from Dr. Joae Wu (University of Massachusetts Medical School). GFP-P53 and FLAG-BCL-2 were from Addgene (Cambridge,MA). pLKO_TRC005 vector was used for silencing PRMT5 (sequence: 5′-AGGGACTGGAATACGCTAATT-3′). pLKO_TRCN000006 2809 Sequence of MSI2 shRNA (sequence:5′CCGGGTGGAAGATGT AAAGCAATATCTCGAGATATTGCTTTACATCTTCCACTTTTTG-3′).

## Xenograft studies

The animals were housed in individually ventilated cages (maximum of 4 animals/cage) with 12 h dark, 12 h light conditions. The animals were fed food and water *ad libitum*. Temperature and relative humidity were maintained at 20 ± 2 °C and 65%, respectively. Female NSG mice (NOD scid gamma mice) were obtained from the Jackson laboratory. 6-week old NSG female mice were injected subcutaneously with 10 million cells together with matrigel. Once tumors reached a volume of 100mm³, mice were randomized to receive either vehicle control (0.5% methylcellulose) or GSK-025, 100 mg/kg orally twice/day for 3 weeks. For the drug combination studies, mice were randomized to receive either vehicle, GSK-025 50 mg/kg twice daily and/or venetoclax 100 mg/kg/daily by oral gavage administrated. Mice were observed daily throughout the treatment period for signs of morbidity/mortality. Tumors were measured three times weekly using calipers, and volume was calculated using the formula: length x width 2 × 0.52. Body weight was also assessed three times weekly. After 3 weeks of treatment tumor samples were collected for immunoblotting. All studies were conducted in accordance with the MSKCC and GSK policies on the Care, Welfare and Treatment of Laboratory Animals and were

reviewed according to the Institutional Animal Care and Use Committee either at MSKCC or GSK. The tumor burden was not exceeded in any of the studies performed.

## MSI2 Hyper-TRIBE

MSI2 Hyper-TRIBE was performed as previously described (29). Z-138 cells were transduced with virus expressing MSI2-ADAR or MIG empty vector. 48 h after transduction GFP-positive cells were sorted. 24 h after treatment with 5uM of GSK-591 and Ro-0812 cells were harvested, and RNA was extracted and sequenced. Identification of RNA editing events was done by following the workflow as previously reported. Essentially, STAR% aligner was used to match paired-end RNA seq reads to the human genome (hg19). The GATK workflow was followed to call variants in RNA-seq to identify all mutations in the RNA-seq libraries. We narrowed focus to those mutations within annotated mRNA transcripts, as well as restricting to solely A-to-G mutations in transcripts encoded by the forward strand and conversely T-to-C mutations in transcripts arising from the reverse strand. Additionally, all the mutations found using the above criteria were filtered using the dbSNP database. We took the sum of the filtered sets of RNA editing events from all RNA-seq libraries within the experiment and totaled the number of reads containing reference (A/T) and alternative (G/C) alleles from each library at each site. Beta-binomial distribution was performed to identify differences of edit frequencies between MIG and MSI2-ADAR, and those conditions with drugs added in the presence of each construct. Significant sites were determined by filtering for FDR-adjusted sites (Benjamini–Hochberg correction), using a *p-value or FDR of* < 0.05.

## Differential expression analyses (DESeq2)

RNA from Z-138 cells treated with 5uM GSK-591 and/or Ro in 3 replicates was extracted with chloroform. Isopropanol and linear acrylamide were added, and the RNA was precipitated with 75% ethanol. Samples were resuspended in RNase-free water. High purity mRNAs were enriched from total RNAs using Dynabeads mRNA purification kit (Thermo Fisher). After PicoGreen quantification and quality control by Agilent BioAnalyzer, mRNA input was used for library preparation (TrueSeq Stranded mRNA LT Sample Prep Kit). Libraries were run on a HiSeq 4000 in a 100 bp/100 bp paired-end run, using the HiSeq 3000/4000 SBS Kit (Illumina). The average number of read pairs per sample was 100 million. Sequence data were aligned using STAR aligner to the human reference genome (version hg19). Fragments Per Kilobase of transcript per Million mapped reads (FPKM) were calculated and differential expression analysis was conducted using the DESeq software package. Differentially expressed genes were identified as those with FPKM greater than 1 showing differential expression greater than twofold (up or down) with an adjusted *p-value* < 0.05.

Raw reads were first analyzed for quality control using FastQC. Reads were then aligned to the hg19 assembly of the human genome using the STAR aligner. The resulting BAM files had their reads counted using the summarizeOverlaps function from the GenomicAlignments package in R. A count table of reads mapping to genes was constructed, and a differential expression analysis was conducted using the DESeq software package. Before significance assessment, genes were filtered according to a reads per kilobase (RPK) value of 500 given the deep average depth of uniquely mapped reads (106, 458, 498) present in the samples. Statistical significance was determined with the negative binomial test using average read counts for DMSO, Ro, GSK-591, and combination for all genes passing the RPK filter. The Benjamini–Hochberg correction was applied to all nominal *P* values to account for multiple hypothesis testing to yield *q* values. Genes were determined to be significantly differentially expressed if they possessed a *q* value that was less than or equal to 0.05.

## Gene pathway enrichment analysis and gene set enrichment analysis

Enriched and depleted genes from the genome-wide CRISPR-Cas9 screen were analyzed for pathway enrichment using ClusterProfiler. DeSeq data and Ro-specific targets were analyzed using the Molecular Signatures Database (MSigDB, 3,284 gene sets) and using the ENRICHR program. For gene set enrichment analysis, we generated rnk files based on genes that were differentially expressed (log2fold change) in response to Ro, GSK591, or combination treatment against the MSigDB and against specific gene sets based on ENRICHR hits. We also generated rnk files for the CRISPR screen hits and ran these against the MSigDB and specific gene sets. Files for each GSEA result are found in the indicated Supplementary data files.

## RNA-immunoprecipitation (RIP)

Z-138 cells were treated with 5uM of GSK-591 and Ro-0812 for 24 h were collected ($20 \times 10^6$ cells were used per IP reaction) and washed twice with ice-cold PBS. Cells were lysed in ice-cold IP lysis buffer (50 mM Tris-HCl pH 7.5, 300 mM NaCl and 0.5% NP40) for 30 min on ice and frozen at −80 °C immediately to aid the lysis. On the day of IP, the lysate was centrifuged to precipitate the debris. Supernatant was collected and incubated with 5 μg of anti-MSI2 antibody (Abcam) or Rabbit IgG (Millipore) overnight at 4 °C. RNA–MSI2 endogenous antibody complexes were pulled down using Dynabeads Protein A/G (Millipore) and washed five times in 100% IP lysis buffer, 70% IP lysis buffer and 30% PBS, 50% IP lysis buffer and 50% PBS, 30% IP lysis buffer and 70% PBS and 100% PBS. RNA was extracted using the phenol–chroloroform method and quantified by qRT–PCR.

## PCR pathway array and qPCR

Total RNA was extracted with the RNeasy mini kit (Qiagen). A total of 1 μg of RNA was converted to cDNA using RT$^2$ First Strand Kit (Qiagen). CFX96 (Bio-Rad Laboratories). mRNA of human oncogenes and tumor suppressor genes were assessed using the PCR pathway array (Qiagen, cat. PAHS-502Z) according to the manufacturer's instructions. Primers for *TP53* (qHsaCED0045022), *c-MYC* (qHsaCID0012921), *BCL-2* (qHsaCED0057245), *SKA2* (qHsaCED0047540), *HEXIM1* (qHsaCED0047867), and *GAPDH* (qHsaCED0038674) were purchased from Bio-Rad Laboratories. Expression of indicated genes were quantified by real-time PCR assays with SYBR Green dye and the CFX96 Touch$^{TM}$ Real-time PCR detection System.

## Luciferase reporter assay

HEK-293T cells were transfected with pRL-CMV MYC 3′UTR, Firefly luciferase control and MSI2 and/or PRMT5 shRNA constructs or either the cells were treated with 5uM of GSK-591 and/or Ro-0812, in the indicated experiments. 48 h post-transfection, expression of Renilla and Firefly luciferase were determined by Dual-Luciferase Reporter Assay System (Promega) following the manufacturer's instructions.

## Apoptosis and cell cycle analysis by flow cytometry

Lymphoma primary cells and cell lines were treated with GSK-591 and Venetoclax. After 72 h, apoptosis and cell death were determined using FITC Annexin V apoptosis detection kit according to the manufacturer's instruction (BD Biosciences, cat. 556547).

To determine the effect of GSK-591 and Ro combination on cell cycle, cell lines were incubated with these compounds for 72 h. $1 \times 10^6$ cells were collected, fixed with 70% ethanol and then stained with propidium iodide (PI)/RNase staining solution (Cell Signaling Technology, cat.4087) at room temperature for 15 minutes. Flow cytometric data were acquired using a FACSCalibur (BD Biosciences, San Jose, CA). Greater than 10,000 events were acquired. Doublets were excluded by gating out high FL3-W (width) cells.

## Immunohistochemistry

Three hours after the last drug treatment, tumors were immediately fixed in fresh 4% paraformaldehyde rotating at 4 °C overnight. Fixed tissues were dehydrated and embedded in paraffin before 5 μm sections were put on slides. The tissue sections were deparaffinized with EZPrep buffer (Ventana Medical Systems) and incubated with antibodies against Ki-67 (DAKO, cat. M7240, 0.5 μg /mL), or c-MYC (Cell signaling, cat. 13987, 4 μg/mL); BCL-2 (Ventana-Roche, cat. 790-4604, 0.2 μg/mL) for 5 h, followed by 60 min of incubation with the corresponding biotinylated secondary antibodies. Slides were scanned with Pannoramic 250 Flash scanner (3DHistech, Hungary) using 20X (0.8 NA) objective lens.

## Statistics

Statistical parameters and analysis are stated in corresponding figure Legends for each panel of data. The Student's $t$ test was performed to determine significance for bar graphs, growth curves, and quantifications unless otherwise specified. Those tests resulting in $p$-values less than 0.05 were deemed significant. Error bars reflect the standard error of the mean (SEM) unless otherwise stated. All statistical analyses were carried out within the programs GraphPad Prism 8.0 and the R statistical environment.

## Reporting summary

Further information on research design is available in the Nature Research Reporting Summary linked to this article.

## Data availability

All datasets generated in this study have been deposited in the Gene Expression Omnibus under the accession number GSE194363. Source data are provided as a Source data file with this manuscript. Source data are provided with this paper.

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

## Acknowledgements

The authors would like to thank the current members of the M. G. Kharas Lab for their critical advice and helpful suggestions. We thank Agnes Viale and members of the Integrated Genomics Operation Core, Katia Manova and members of MSKCC Molecular Cytology Core. This work was supported in part by the NIH MSK SPORE in Lymphoma (P50 CA192937), Leukemia & Lymphoma Society Specialized Center of Research Program (7014-17), Steven A Greenberg Research Grant, and the George Ohrstrom Foundation. A.Y. M.G.K. is a Scholar of the Leukemia and Lymphoma Society and supported by NIDDK NIH R01-DK101989-01A1, NCI 1R01CA193842-01, NCI 1R01CA193842-06A1, 5R01CA186702-07, 1R01DK1010989-06A1, R01HL135564, and R01CA 225231-01; The Starr Cancer Consortium; the Alex's Lemonade Stand A Award. C.M.E. is supported by NCI 5 F31 CA257204-02. MSK core facilities are supported by P30CA008748.

## Author contributions

T.E. led this project, designed, and performed experiments, analyzed data, and wrote the manuscript. C.M.E. performed and analyzed experiments, and helped prepare the manuscript. D.Z.; A.Y.R.; Z.A.; M.D.S.; S.M.; and X.P.Z. all performed experiments and analyzed data. E.L.C; X.Y.; M.V.R.; M.F.; and C.S.L. performed and supervised bioinformatic analysis. A.K.; V.S. performed statistical analysis. C.L.B. provided clinical data and analysis. R.G.; E.D.S.; O.B.; and A.M.M. provided critical reagents and suggestions. A.Y.; M.G.K. directed the project, analyzed data, and wrote the manuscript.

## Competing interests

A.Y. is currently employed by AstraZeneca. Research support: Janssen, Curis, Merck, BMS, Syndax, Roche; Honorarium: Janssen, Abbvie, Merck, Curis, Epizyme, Roche, Takeda; Consulting: Biopath, Xynomics, Epizyme, Roche, Celgene, HCM. C.L.B. Research Support: Janssen, Novartis, Epizyme, Bayer, Autolus, Roche; Honorarium: Dava Oncology, TouchIME, Medscape. Consulting: Life Sci, GLG, Juno/Celgene/BMS, Seattle Genetics, Kite, Karyopharm, TG Therapeutics, ADC Therapeutics.A.M.M. Research support: Janssen Pharmaceuticals, Sanofi, and Daiichi Sankyo. Consulting: Epizyme and Constellation. Scientific advisory board of KDAC pharmaceuticals. O.B. is currently employed by GlaxoSmithKline. M.G.K. honorarium at Kumquat Biosciences and Scientific advisory board 858 Therapeutics. The remaining authors declare no competing interests.
