## [Peer Review File · Nature Communications]

REVIEWER COMMENTS

Reviewer #1 (Remarks to the Author):

This manuscript by Erazo and colleagues reports on novel mechanisms conferring resistance and sensitivity to a PRMT5 inhibitor in clinical evaluation in different models of B-cell lymphoma. They found TP53 and MSI2 as sensitizing and resistant drivers to the inhibitor, respectively. By genetic and pharmacological experiments, they show that MSI2 depletion enhanced GSK-591 anti-lymphoma effect. Interestingly, MSI2 and PRMT5 inhibition blocked c-MYC and BCL-2 translation, the latter sensitizing to BCL-2 depletion or inhibition with venetoclax in vitro and in vivo. Based on these results, a therapeutic strategy combining the PRMT5 inhibitor with MSI2 or BCL-2 inhibition is proposed for lymphomas.

This is a well written and elegantly conducted work, which raises interesting scientific conclusions with potential clinical impact. A large number of experiments support the conclusions of the paper, but I have major and minor concerns that in my opinion will improve the quality of this paper.

Major

The recurrent P53 mutation at R248W is a well-known change inducing P53 gain of function in many cancer types, although in terms of GSK-591 responses in lymphoma, it drives similar results to that of P53 deletion. Is this because this P53 change behaves differently in lymphoma in comparison to other tumors, or because R248W is in reality a loss-of-function mutation?

The two selected lymphoma cell lines used in the initial CRISPR genetic screen exhibit normal function of P53, and thus yielded to the discovery of MSI2 as a mediator of GSK-591 resistance. This P53-MSI2 link is well demonstrated in in vitro studies in the cell lines, but I have not clear whether in clinical samples (i.e. in a clinical trial testing GSK-591 or other similar inhibitors in lymphoma patients), those with P53 inactivation would be resistant to the drug, or therapeutic resistance would be dictated by P53 inactivation and the levels of expression of MSI2. In this line, can authors estimate the percentage of patient with lymphoma that will respond to the drug according to the data presented here?

Because of the novelty of the role of MSI2 in lymphoma, the expression of this gene alone and together with PRMT5 should be evaluated in the very many series of lymphoma patients available, according to the type of lymphomas according to the current classifications, and the subsequent subclassifications (DLBCL and novel genetic subtypes, MCL and clinic-pathologic subtypes, etc.). Available data including survival would be also interesting, in addition to Fig.3C. Data from CLL patients can also be included, as the drug could also be of value in this disease. Moreover, MSI2 expression along normal B-cell differentiation, including germinal center stages, would be helpful in order to understand its potential role in lymphoma.

Minor

Introduction: among the different oncogenic functions and targets/partners of PRMT5 in B-cell lymphomas (P53, MYC, Cyclin D1), one that links PRMT5 and BCL6 in regulating the germinal center function is missing (ref.21).

In Fig.2A, PARP cleavage is not clearly increased in both Z138 and OCI-LY19 cells. Indeed, in Z138 NT cells, both total and cleaved forms of the protein are increased, is there a proportional increase with respect to the KO cells? In addition, PARP cleavage is not explored in the following Fig.2C. Is there a reason?

Where does data in Fig.3A (DLBCL patients) come from? As mentioned before, this section requires substantial expansion.

Figure 3B could be improved by adding a Pearson correlation study to correlate MSI2 and PRMT5 expression levels.

Figure 3H requires quantification and statistical evaluation.

What are the differences in GSEA in Figures 4F and 4G, related to MYC target genes?

In Figure 4I, synergistic treatment increases MDM2 and P53. Because MDM2 is normally a P53 repressor, do the authors have an explanation to this finding?

What is the scientific rationale for mixing data from the three studies resulting in Figure 7A? It is clear that this way the authors picked 3 genes including BCL2, but I just wonder whether finding BCL2 as a target gene in the initial experiments from the CRISPR screens would be good enough, given the biological significance of BCL2, and because venetoclax is available for lymphoma therapeutics.

Reviewer #2 (Remarks to the Author):

In this study Erazo and colleagues set out to identify genetic drivers of sensitivity and resistance to the PRMT5 inhibitor GSK-591 using CRISPR/Cas9 screening. They identify that TP53 loss/mutation confers resistance to GSK-591, while targeting of MSI2 genetically or with Ro pharmacologically cooperates with GSK-591 to reduce cell growth. They uncover the gene expression pathways perturbed by Ro and PRMT5 inhibition. Furthermore, they identify the impact of combined MSI2 and PRMT5 inhibition on BCL2 translation to characterise the impact of the BCL2 inhibitor venetoclax together with PRMT5 inhibition on tumor cell growth. They propose that combined PRMT5 and MSI2/BCL2 inhibition could be an effective therapeutic strategy.

Overall the findings are novel with the potential to influence the clinical use of PRMT5 inhibitors. The work is clearly presented and the data broadly support the claims of the study. However, there are some points which should be addressed prior to further consideration of publication:

1. Why did the authors select Z-138 cells for their screen?
2. The authors should explain the analytic methods used to identify sensitizing and resistance genes in greater detail.
3. Why did the authors select OCI-LY19 cells for validation purposes? Did they validate in the original Z-138 cells or other cell lines?
4. Did the authors conduct a screen in any of their GSK-591-resistant cells to identify sensitizing genes?
5. Why did only 32/316 sensitizing and 4/89 resistance genes validate in OCI-LY19 cells? What are the characteristics of the genes which did not validate?
6. The authors associate TP53 mutations and MSI2 overexpression with resistance to GSK-591. Do these two features account for GSK-591 resistance in the 14 cell lines with high IC50s? If not, can the authors speculate why not? Some of the drug-sensitive lines also appear to carry TP53 mutations. An immunoblot of MSI2 levels across cell lines in Fig 1B would be useful in addition to Fig. 3D.
7. Fig 3J - absolute values should also be presented in addition to relative values.
8. The authors overlapped three datasets 'to identify MSI2 targets in lymphoma' in Fig 7A leading the identification of BCL2 and SKA2. The rationale behind overlapping these three datasets is not clear and the authors should elaborate their approach further.

Reviewer #3 (Remarks to the Author):

In this paper, authors found that the resistance to PRMT5 inhibitor GSK-591 in B-cell lymphoma can be predicted by TP53 gene deletion or the TP53 R248W mutation. In addition, MSI2 and its downstream targets c-MYC and BCL-2 were found to be responsible for the resistance to GSK-591.

The authors provided novel drug combination strategies by targeting these gene products in B-cell lymphoma. Although there are some missing links in the action mechanism, these conclusions have clinical implication to better use of PRMT5 inhibitors. In general, the work is conceptually interesting and the data support their claims well. However, several concerns should be addressed.

Main concerns:

1. The western blotting data which are related to the main conclusions of article should provide quantification bar graphs and statistics.
2. In the Fig. 6C, hyperTRIBE experiment should include ADAcd_only as a background control group. Different experimental conditions in hyperTRIBE may give rise to different level of noise. The edited sites detected in ADAcd_only should be removed from those in MSI2-ADA to reduce noise.

Minor concerns:

1. In the line 98-99 and the line 260-261, it would be better to cite the corresponding reference to claim TP53 is a well characterized target of PRMT5 and to introduce Ki-67 respectively. In the line 721, the reference is not correct.
2. In Fig. 1B, OCI-LY19 was misspelled as 'OCY-LY19'. In the line 418, 'Supplementary Fig. 5B' should be substituted by 'Supplementary Fig. 6B'.
3. In Fig. 2G, statistical significance is indicated by ****, while *** $p < 0.001$ is written in the legend. What is the statistical method of the experiment? In Supplementary Fig. 4E and Supplementary Fig. 6I, how many experiments were carried out? The 'n' number and statistical method should be mentioned. Also in Supplementary Fig. 6I, the sample names in the FACS picture and the bar graph are not consistent. 'Ro' in the bar graph looks incorrect.
4. The resolution of Fig. 7 is very low, and the fonts of characters in some Figures are too small to see.

Response to Reviewers

Reviewer # 1

This manuscript by Erazo and colleagues reports on novel mechanisms conferring resistance and sensitivity to a PRMT5 inhibitor in clinical evaluation in different models of B-cell lymphoma. They found TP53 and MSI2 as sensitizing and resistant drivers to the inhibitor, respectively. By genetic and pharmacological experiments, they show that MSI2 depletion enhanced GSK-591 anti-lymphoma effect. Interestingly, MSI2 and PRMT5 inhibition blocked c-MYC and BCL-2 translation, the latter sensitizing to BCL-2 depletion or inhibition with venetoclax *in vitro* and *in vivo*. Based on these results, a therapeutic strategy combining the PRMT5 inhibitor with MSI2 or BCL-2 inhibition is proposed for lymphomas.

This is a well written and elegantly conducted work, which raises interesting scientific conclusions with potential clinical impact. A large number of experiments support the conclusions of the paper, but I have major and minor concerns that in my opinion will improve the quality of this paper.

Major

1. The recurrent P53 mutation at R248W is a well-known change inducing P53 gain of function in many cancer types, although in terms of GSK-591 responses in lymphoma, it drives similar results to that of P53 deletion. Is this because this P53 change behaves differently in lymphoma in comparison to other tumors, or because R248W is in reality a loss-of-function mutation?

Response: The reviewer raised an interesting question as there is some controversy whether *TP53* missense mutations result in gain or loss of function. Based on previous studies the *TP53* mutations that are localized in the DBD are likely to result in a loss of *TP53* transcriptional activity¹. We demonstrated in Supplemental Fig. 2C and Fig. 2F that R248W mutation impaired the activation of P53 targets on the mRNA and protein levels, respectively. Additionally, in accordance with *in vitro* and *in vivo* models and AML primary samples *TP53* missense mutations, including mutations in codon R248, exhibited dominant-negative activity without evidence of GOF activity². Together, these data strongly support that *TP53* loss of function activity is not limited to lymphoma.

2. The two selected lymphoma cell lines used in the initial CRISPR genetic screen exhibit normal function of P53, and thus yielded to the discovery of MSI2 as a mediator of GSK-591 resistance. This P53-MSI2 link is well demonstrated in *in vitro* studies in the cell lines, but I have not clear whether in clinical samples (i.e. in a clinical trial testing GSK-591 or other similar inhibitors in lymphoma patients), those with P53 inactivation would be resistant to the drug, or therapeutic resistance would be dictated by P53 inactivation and the levels of expression of

MSI2. In this line, can authors estimate the percentage of patient with lymphoma that will response to the drug according to the data presented here?

Response: We thank the reviewer for this important comment and we have now more carefully examined the relationship between MSI2 and *TP53*. In the revised Figure 3E, we now compare the IC50s of GSK-591 and MSI2 abundance determined by immunoblot and further subdivide the cell lines based on *TP53* mutational status. We found that either high MSI2 abundance or *TP53 wt* was significantly associated with a lower IC50 of GSK-591. Additionally, MSI2 high and *TP53* mutant cells lines were not significantly associated with sensitivity to PRMT5 inhibition. Therefore, we proposed that together MSI2-high and *TP53 wt* are predictors of response to PRMT5 inhibitors (Response Figure 1A).

Since the results from the Phase I study with the GSK591 analogue, GSK3326595, in Non-Hodgkin's lymphomas (NCT02783300) have not been published, we were not able to determine the MSI2 status and the clinical response in this study. Alternatively, we analyzed *MSI2* expression and *TP53* genomic status in an RNA-Seq dataset (Staudt), a cohort of 480 *de novo* ABC and GCB-DLBCL cases³. There is no enrichment for MSI2 or *TP53* status in this cohort (Fisher's Test 0.75). Based on the cell line data, we predict that 38% (183/480) of the patients from this cohort would respond to PRMT5 inhibitors, as they meet the criteria of high *MSI2* expression and *TP53 wt* (Response Figure 1B). Overall, in the revised manuscript, we provide this new data suggesting that MSI2 and *TP53* status correlates to PRMT5 inhibition.

AMSI2 expression vs
GSK-591 response*TP53* status vs
GSK-591 responseMSI2 / *TP53* vs GSK-591 response**B**Correlation *MSI2* expression with *TP53*
genomic status in DLBCL patients

N = 480		
Characteristic	TP53 wt	TP53 mutant
MSI2 low	175 (36.5%)	57 (11.87%)
MSI2 high	183 (38.1%)	65 (13.5%)

Response Figure 1. Correlation of *MSI2* expression and *TP53* mutation status. **A.** Bar graph showing IC₅₀ values of GSK-591 after 8 days of treatment, *MSI2* protein abundance analyzed by immunoblot and *TP53* mutation status in 16 cell lines from different lymphoma subtypes. **Left.** Dot plot comparing IC₅₀ values of GSK-591 in cells with low *MSI2* expression (below *MSI2*/*ACTIN* ratio) and high *MSI2* (above *MSI2*/*ACTIN* ratio). **Right.** Dot plot comparing IC₅₀ values of GSK-591 in cells *TP53* wt and mutant. **Bottom:** Dot plot comparing IC₅₀ values of GSK-591 in the indicated conditions. *p*-values were determined using one-tailed *t* test. **B.** Table showing the number of DLBCL patients and percentages in the indicated conditions from the Staudt RNA-Seq dataset. *MSI2* mRNA expression was dichotomized at median.

- Because of the novelty of the role of *MSI2* in lymphoma, the expression of this gene alone and together with *PRMT5* should be evaluated in the very many series of lymphoma patients available, according to the type of lymphomas according to the current classifications, and the subsequent subclassifications (DLBCL and novel genetic subtypes, MCL and clinic-pathologic subtypes, etc.). Available data including survival would be also interesting, in addition to Fig.3C. Data from CLL patients can also be included, as the drug could also be of value in this disease. Moreover, *MSI2* expression along normal B-cell differentiation, including germinal center stages, would be helpful in order to understand its potential role in lymphoma.

Response: We thank the reviewer for suggesting us to expand our analysis of *MSI2* in different lymphoma patient datasets and that could further clarify its role in lymphoma. We first examined the expression of *MSI2* across different DLBCL subtypes ABC, GCB and other non-characterized subtypes from BC Cancer Lymphoid Cancer Dataset (n=322). We observed an increased in *MSI2* expression in GCB-DLBCL (Response Fig. 2A). We evaluated if there is correlation between *PRMT5* and *MSI2* expression. For that, we analyzed the expression of these two genes in two DLBCL RNA-Seq datasets. BCCA dataset, a cohort of 322 *de novo* ABC and GCB-DLBCL cases, all treated with rituximab plus CHOP at the BC Cancer Agency (Vancouver)⁴ and Staudt dataset. The Pearson correlation analysis in the new Figure 3B, shows that *PRMT5* expression correlates positively with *MSI2* expression in ABC and GCB-DLBCL patients (n= 820) (Response Fig. 2B). As the reviewer pointed out, previously it has been shown that high *MSI2* expression had shorter Overall Survival (OS) in CLL patients⁵ (Response Fig. 2C). In contrast to lymphoma patients, *PRMT5* and *MSI2* expression are not correlated in this cohort of 156 CLL patients⁶ (Response Figure 2D).

Next, we evaluated whether *MSI2* is a biomarker of poor survival in three cohorts of DLBCL patients, BCCA, Staudt and a microarray dataset (accession number GSE10846) of a cohort of 414 *de novo* DLBCL, of which 181 patients were treated with CHOP and 233 patients with Rituximab-CHOP⁷. Response Fig. 2E-F are showing that even though *PRMT5* was considered a biomarker of poor prognosis in DLBCL^{8,9}, in two of the analyzed datasets, BCCA and Staudt, its expression was not significantly associated with OS. Also, high *MSI2* expression was not found associated with poor prognosis in the 1234 DLBCL patients from the three cohorts (Response Figure 2G).

To characterize the requirement for *Msi2* in normal B-cell development and germinal center formation, we performed an analysis of *Msi2* mRNA levels in the major B-cell populations. Gene expression analysis (Immgen) indicates that *Msi2* expression is expressed across all B-cell subsets. However, *Msi2* expression is higher in B-cell progenitors from the bone marrow and slightly decays through the maturation process. B-cells from the marginal zone and peritoneal cavity expressed low *Msi2* levels. These data suggested that *Msi2* may contribute to normal germinal center formation and this may provide insight into the cell of origin and development of B-cell lymphomagenesis (Response Fig. 2H).

In summary, in the revised manuscript, we incorporated the new data describing that *MSI2* is highly expressed in GCB-DLBCL and its expression strongly correlates with *PRMT5* in DLBCL patients. We moved the original Figure 3C to the supplementary data section and state in the results section that in a smaller dataset high *MSI2* predicts a poor outcome (TCGA; n=47) but could not be validated as an independent marker in larger datasets.

Response Figure 2. *MSI2* and *PRMT5* strongly correlate in lymphoma patients. **A.** Bar graph showing *MSI2* levels (FKPM) in ABC, GCB-DLBCL and non-characterized subtypes from the BC Cancer Lymphoid Cancer Database (n=322). **B.** *PRMT5* is positively correlated with *MSI2* expression. Pearson correlation coefficient of *PRMT5* and *MSI2* expression across 820 DLBCL patients. r (95%) = 0.588 p -value < 0.0001. **C.** Kaplan–Meier curve showing that high *MSI2* mRNA levels predicts poor OS in CLL patients (Figure from Palacios F., et al. 2021. *Leukemia*). **D.** Pearson correlation coefficient of *PRMT5* and *MSI2* expression in a cohort of 156 CLL patients. r (95%) = 0.07 p -value = 0.387. **E-F.** Analysis of *MSI2* and *PRMT5* expression versus overall survival (OS) in BCCA dataset (n= 322) and Staudt (n= 480), respectively. Expression of *MSI2* and *PRMT5* were dichotomized at median and the Cox regression analysis was performed to assess the clinical association between these variables. Neither *PRMT5* nor *MSI2* are statistically significantly associated with OS. HR = Hazard Ratio, CI= Confidence Interval. **G.** Kaplan-Meier curves indicating that high *PRMT5* expression (left) is statistically significantly associated with poor survival and *MSI2* expression (right) is not significantly associated with clinical outcome in a cohort of 414 DLBCL patients. **H.** *Msi2* expression in the major B-cell subsets (Immunological Genome Project, Immgen): bone marrow; fetal liver; spleen; mesenteric lymph node; lymph node; peritoneal cavity. ***comparison of common lymphoid progenitor from the bone marrow vs B-cells from the peritoneal cavity.

Minor

- Introduction: among the different oncogenic functions and targets/partners of PRMT5 in B-cell lymphomas (P53, MYC, Cyclin D1), one that links PRMT5 and BCL6 in regulating the germinal center function is missing (ref.21).

Response: We added this reference to the line 85 of the introduction.

“Moreover, PRMT5 cooperates with MYC to sustain splicing fidelity which is key to ensure tumor maintenance of MYC-driven lymphomas⁹ and promotes the upregulation of BCL-6 repressive activity in DLBCL¹²”

- In Fig.2A, PARP cleavage is not clearly increased in both Z138 and OCI-LY19 cells. Indeed, in Z138 NT cells, both total and cleaved forms of the protein are increased, is there a proportional increase with respect to the KO cells? In addition, PARP cleavage is not explored in the following Fig.2C. Is there a reason?

Response: To confirm the significant increase in PARP cleavage in NT cells (control) with respect to the KO cells, we are showing in the revised version of the manuscript the quantification of the western blots in Fig, 2A. As suggested, we analyzed PARP cleavage and added the western blot to the new Fig. 2C.

Response Figure 3. Mino and Rec-1 GSK-591-resistant cell lines were transfected with 1 μg of GFP-P53 and 2 h after transfection were treated with GSK-591 (5 μM). The effect on PARP expression was analyzed by immunoblot.

- Where does data in Fig.3A (DLBCL patients) come from? As mentioned before, this section requires substantial expansion.

Response: We agree with the reviewer that this section required further exploration. The data from Fig. 3A is from TCGA database in GEPIA, that it is a cohort of DLBCL patients (n= 47) versus normal B-cells (n=337) from GTEX.

- Figure 3B could be improved by adding a Pearson correlation study to correlate MSI2 and PRMT5 expression levels.

Response: As suggested, we performed a Pearson correlation analysis, showed in the new Fig. 3C and described in the response # 3.

8. Figure 3H requires quantification and statistical evaluation.

Response: As suggested, we quantified Ki-67 expression in the different conditions from the *in vivo* study and added a bar graph in the new Fig. 3I.

Response Figure 4. Bar graphs representing the percentage of total sectional area positive for IHC-based staining of Ki-67 in Z138-derived xenografts of the indicated treatment groups.

9. What are the differences in GSEA in Figures 4F and 4G, related to MYC target genes?

Response: GSEA in Fig. 4F and 4G are demonstrating the overall downregulation of MYC target gene signature in the combination of Ro and GSK581 with respect to either Ro alone in Fig. 4F or the comparison with GSK-591 alone in 4G. These data indicate that the loss of the MYC signature is significantly enriched with combined treatment comparing the solo treatment conditions.

10. In Figure 4I, synergistic treatment increases MDM2 and P53. Because MDM2 is normally a P53 repressor, do the authors have an explanation to this finding?

Response: The reviewer raises here an interesting point. As the reviewer pointed out, it is well established that MDM2 has E3 ubiquitin ligase activity and targets P53 for proteasomal degradation. However, it has been also shown that MDM2 is a target of P53 and they form a negative-feedback loop, in which P53 induces the expression of MDM2, which in turn represses P53 activity and promotes its degradation^{10, 11}. Thus, it is not surprising that in our study the upregulation of P53 results in induction of MDM2 and other canonical P53 target genes including P21.

11. What is the scientific rationale for mixing data from the three studies resulting in Figure 7A? It is clear that this way the authors picked 3 genes including BCL2, but I just wonder whether finding BCL2 as a target gene in the initial experiments from the CRISPR screens would be good enough, given the biological significance of BCL2, and because venetoclax is available for lymphoma therapeutics.

Response: Our goal for the overlap of the three different datasets was to identify MSI2-specific targets that could play a role in resistance to PRMT5 inhibition. By analyzing only CRISPR screen hits, we could have missed the identification of the novel PRMT5/MSI2/c-

MYC/BCL-2 axis. In the revised manuscript, we added the justification for this overlap (line 403).

Reviewer #2

In this study Erazo and colleagues set out to identify genetic drivers of sensitivity and resistance to the PRMT5 inhibitor GSK-591 using CRISPR/Cas9 screening. They identify that TP53 loss/mutation confers resistance to GSK-591, while targeting of MSI2 genetically or with Ro pharmacologically cooperates with GSK-591 to reduce cell growth. They uncover the gene expression pathways perturbed by Ro and PRMT5 inhibition. Furthermore, they identify the impact of combined MSI2 and PRMT5 inhibition on BCL2 translation to characterise the impact of the BCL2 inhibitor venetoclax together with PRMT5 inhibition on tumor cell growth.

They propose that combined PRMT5 and MSI2/BCL2 inhibition could be an effective therapeutic strategy. Overall the findings are novel with the potential to influence the clinical use of PRMT5 inhibitors.

The work is clearly presented, and the data broadly support the claims of the study. However, there are some points which should be addressed prior to further consideration of publication:

12. Why did the authors select Z-138 cells for their screen?

Response: We focused our study in a Mantle Cell Lymphoma model as this is the subtype of lymphoma with the worst clinical outcome and novel targeted therapies are needed¹². It has been reported that PRMT5 is overexpressed in MCL^{13,14}, so it was interesting to evaluate the therapeutic value of its inhibition in this disease. Moreover, MCL cell line, Z-138, is one of the most sensitive cell lines to GSK-591.

13. The authors should explain the analytic methods used to identify sensitizing and resistance genes in greater detail.

Response: As requested, we have added a paragraph in the methods in the line 637, describing the analytical methods used for hit identification. "We applied both the differential expression (adjust *p-value* <0.1 and \log_2 FoldChange < -2 for resistance genes and \log_2 FoldChange > 2 for sensitizing genes) and the abundance of sgRNAs in the same direction, depletion or enrichment (number of sgRNAs per gene \geq 3) criteria to screen for differential sgRNAs"

14. Why did the authors select OCI-LY19 cells for validation purposes? Did they validate in the original Z-138 cells or other cell lines?

Response: We decided to perform the validation screen in a DLBCL model because we wanted to establish if the mechanisms of resistance and sensitivity to GSK-591 identified in MCL, were consistent across different lymphoma subtypes. We chose the DLBCL cell line, OCI-LY19, as it is also one of the most sensitive cells to GSK-591 with high lentiviral transduction efficacy.

In addition to the second screen, we used several methods to validate the top-ranked hits, *TP53* and *MSI2*, individually. Thus, we confirmed the role of P53 in sensitivity to GSK-591 by generating isogenic cell lines, using sgRNAs targeting P53, in Z-138 and OCI-LY19 cells (Fig. 2A and B). We also found that P53 wt overexpression in mutant cell lines rescues the resistance to GSK-591 (Fig. 2D).

To validate *MSI2* as a top driver of resistance, we confirmed the effect of *MSI2* genetic depletion using a short hairpin (shRNA) targeting *MSI2* in four cell lines from different lymphoma subtypes, in the two sensitive, Z-138 and OCI-LY19, and two resistant cell lines, CA-46 and HBL-1 (Fig. 3G and Supplementary Fig. 3C, respectively). Moreover, we performed rescue experiments, overexpressing *MSI2* in Z-138 and OCI-LY19 cells and confirmed that high *MSI2* expression confers resistance to PRMT5 inhibitor (Fig. 3F).

15. Did the authors conduct a screen in any of their GSK-591-resistant cells to identify sensitizing genes?

Response: We did not conduct a validation screen in a GSK-591-resistant cell line, because the doses of GSK-591 necessary to kill above 80% of resistant cells are high (> 30uM) that could lead to off-target effects. This could produce false negatives, decreasing the confidence of the biological relevance of the hits.

16. Why did only 32/316 sensitizing and 4/89 resistance genes validate in OCI-LY19 cells?

Response: The reasons that could explain why only 36 hits from Z-138 CRISPR screen were validated in the OCI-LY19 screen. First, these two cell lines belong to different subtypes of lymphoma, MCL and DLBCL, so they have different genetic backgrounds, which would explain why some drug targets are different in these cell lines. Second, the selection in both screens was very stringent (FDR < 0.05 and \log_2 FoldChange < -2 for resistance genes and \log_2 FoldChange > 2 for sensitizing genes) to ensure we identify only the genes with the strongest effect on the GSK-591 activity.

17. What are the characteristics of the genes which did not validate?

Response: A gene set enrichment analysis using the Molecular Signatures Database (MSigDB) from the ENRICH program showed that the 360 genes not validated in the second screen are enriched for mitochondrial and ribosomal biogenesis programs. These findings increase the confidence that the genes that were validated in OCI-LY19 represent are true positives, that are potential drug targets relevant for GSK-591 activity in different lymphoma subtypes.

Response Figure 5. Top significant enriched Molecular Signatures (MSigDB) in the non-validated hits from the OCI-LY19 screening using ENRICH analysis.

18. The authors associate TP53 mutations and MSI2 overexpression with resistance to GSK-591. Do these two features account for GSK-581 resistance in the 14 cell lines with high IC50s? If not, can the authors speculate why not? Some of the drug-sensitive lines also appear to carry TP53 mutations.

Response: The reviewer raised a very interesting point. We addressed this by providing new data in Reviewer's comment # 2. New Figure 3E is showing that together, *TP53* wt and high MSI2 expression are biomarkers of response to PRMT5 inhibitors in lymphoma cell lines.

19. An immunoblot of MSI2 levels across cell lines in Fig 1B would be useful in addition to Fig. 3D.

Response: We agree with the reviewer that adding an immunoblot of MSI2 levels across cell lines could be very informative. We added this data now in the Fig. 3E (described in the response #2), instead of Fig. 1B.

20. Fig 3J - absolute values should also be presented in addition to relative values.

Response: We added a bar graft showing the absolute values in Fig. 3J, as requested.

Response Figure 6. Combination of Ro and GSK-591 is synergistic in lymphoma cell lines. Bar graph showing the absolute number of viable cells in three different cell lines treated with 10uM of each agent and the combination. Cell viability was monitored by Celltiter-Glo assay 72 h after drug treatment.

21. The authors overlapped three datasets ‘to identify MSI2 targets in lymphoma’ in Fig 7A leading the identification of BCL2 and SKA2. The rationale behind overlapping these three datasets is not clear and the authors should elaborate their approach further.

Response: We added the following sentence to the line 403, “to identify additional MSI2 direct targets that promote resistance to PRMT5 inhibition in B-cell lymphoma, we overlapped.”.

Reviewer #3

In this paper, authors found that the resistance to PRMT5 inhibitor GSK-591 in B-cell lymphoma can be predicted by TP53 gene deletion or the TP53 R248W mutation. In addition, MSI2 and its downstream targets c-MYC and BCL-2 were found to be responsible for the resistance to GSK-591. The authors provided novel drug combination strategies by targeting these gene products in B-cell lymphoma. Although there are some missing links in the action mechanism, these conclusions have clinical implication to better use of PRMT5 inhibitors. In general, the work is conceptually interesting and the data support their claims well. However, several concerns should be addressed.

Main concerns:

22. The western blotting data which are related to the main conclusions of article should provide quantification bar graphs and statistics.

Response: We added the quantification of western blots in Fig. 2A, 3D, 5B and 7C.

23. In the Fig. 6C, hyperTRIBE experiment should include ADACd_only as a background control group. Different experimental conditions in hyperTRIBE may give rise to different level of noise. The edited sites detected in ADACd only should be removed from those in MSI2-ADA to reduce noise.

Response: Previously, our group demonstrated that there are not significant differences in edit frequency between MIGR1 control and MSI2 catalytic dead domain, thus they both can serve as controls of the HyperTRIBE experiment¹⁵.

Minor concerns

24. In the line 98-99 and the line 260-261, it would be better to cite the corresponding reference to claim TP53 is a well characterized target of PRMT5 and to introduce Ki-67 respectively. In the line 721, the reference is not correct.

Response: We added the references to the line 96. As suggested we introduced Ki-67 staining. By adding the following paragraph in the line 263: “We confirmed the synergistic effect of PRMT5 inhibition and MSI2 knockdown on cell proliferation by performing the immunohistochemical analysis of Ki-67, a biomarker of proliferating cancer cells²⁵”.

25. In Fig. 1B, OCI-LY19 was misspelled as ‘OCY-LY19’. In the line 418, ‘Supplementary Fig. 5B’ should be substituted by ‘Supplementary Fig. 6B’.

Response: We corrected the text accordingly.

26. In Fig. 2G, statistical significance is indicated by ****, while *** $p < 0.001$ is written in the legend. What is the statistical method of the experiment? In Supplementary Fig. 4E and Supplementary Fig. 6I, how many experiments were carried out? The ‘n’ number and statistical method should be mentioned. Also in Supplementary Fig. 6I, the sample names in the FACS picture and the bar graph are not consistent. ‘Ro’ in the bar graph looks incorrect.

Response: The statistical method used was one-way ANOVA. We corrected the figure legend in the line 1110 “different experiments **** $p < 0.0001$ ”. The experiments in Supplementary Fig. 4E and Supplementary Fig. 6I were performed in triplicates and statistical method used was two-tailed t test, we have included this information in the figure legends, lines 92 and 140, respectively.
We have corrected the title in the bar graph, replacing Ro for venetoclax in Supplementary Fig. 6I.

27. The resolution of Fig. 7 is very low, and the fonts of characters in some Figures are too small to see.

Response: We increased the fonts in Fig. 7I, 7H, I and K. We have increased the resolution of the Fig. 7 in the revised manuscript.

References

1. Olivier M, Hollstein M, Hainaut P. TP53 mutations in human cancers: origins, consequences, and clinical use. *Cold Spring Harb Perspect Biol* **2**, a001008 (2010).

2. Boettcher S, *et al.* A dominant-negative effect drives selection of TP53 missense mutations in myeloid malignancies. *Science* **365**, 599-604 (2019).
3. Schmitz R, *et al.* Genetics and Pathogenesis of Diffuse Large B-Cell Lymphoma. *N Engl J Med* **378**, 1396-1407 (2018).
4. Ortega-Molina A, *et al.* The histone lysine methyltransferase KMT2D sustains a gene expression program that represses B cell lymphoma development. *Nat Med* **21**, 1199-1208 (2015).
5. Palacios F, *et al.* Musashi 2 influences chronic lymphocytic leukemia cell survival and growth making it a potential therapeutic target. *Leukemia* **35**, 1037-1052 (2021).
6. Landau DA, *et al.* Mutations driving CLL and their evolution in progression and relapse. *Nature* **526**, 525-530 (2015).
7. Lenz G, *et al.* Stromal gene signatures in large-B-cell lymphomas. *N Engl J Med* **359**, 2313-2323 (2008).
8. Szablewski V, *et al.* An epigenetic regulator-related score (EpiScore) predicts survival in patients with diffuse large B cell lymphoma and identifies patients who may benefit from epigenetic therapy. *Oncotarget* **9**, 19079-19099 (2018).
9. Koh CM, *et al.* MYC regulates the core pre-mRNA splicing machinery as an essential step in lymphomagenesis. *Nature* **523**, 96-100 (2015).
10. Moll UM, Petrenko O. The MDM2-p53 interaction. *Mol Cancer Res* **1**, 1001-1008 (2003).
11. Nag S, Qin J, Srivenugopal KS, Wang M, Zhang R. The MDM2-p53 pathway revisited. *J Biomed Res* **27**, 254-271 (2013).
12. Jain P, Dreyling M, Seymour JF, Wang M. High-Risk Mantle Cell Lymphoma: Definition, Current Challenges, and Management. *J Clin Oncol* **38**, 4302-4316 (2020).
13. Chung J, Karkhanis V, Baiocchi RA, Sif S. Protein Arginine Methyltransferase 5 (PRMT5) Promotes Survival of Lymphoma Cells via Activation of WNT/beta-CATENIN and AKT/GSK3beta Proliferative Signaling. *J Biol Chem*, (2019).
14. Chan-Penebre E, *et al.* A selective inhibitor of PRMT5 with in vivo and in vitro potency in MCL models. *Nat Chem Biol* **11**, 432-437 (2015).
15. Nguyen DTT, *et al.* HyperTRIBE uncovers increased MUSASHI-2 RNA binding activity and differential regulation in leukemic stem cells. *Nat Commun* **11**, 2026 (2020).

REVIEWERS' COMMENTS

Reviewer #2 (Remarks to the Author):

The authors have addressed my queries; I have no other concerns.

Reviewer #3 (Remarks to the Author):

The revised version of manuscript is improved very much. However, I don't think they have responded the 23th comment correctly. And there are some figures still with low resolution.

Response to Reviewers

Reviewer #1

Reviewer #1 had no comments after first resubmission. We thank Reviewer #1 for their time and input to improve this manuscript.

Reviewer # 2

“The authors have addressed my queries; I have no other concerns.”

Reviewer #2 had no comments after first resubmission. We would like to thank Reviewer 2 for their time and input to improve this manuscript.

Reviewer #3

“The revised version of manuscript is improved very much. However, I don’t think they have responded the 23th comment correctly. And there are some figures still with low resolution.”

We thank Reviewer #3 for their comments saying the manuscript has been improved.

To address the concern with the 23rd point from the first rebuttal letter, Reviewer #3 had previously asked us to repeat the experiments and all downstream analysis including a vector containing a catalytically dead ADAR. In our lab’s previous work, Nguyen et al. *Nat. Comms*, 2020, it was shown that the ADAR editing activity detected between empty vector (MIGR1) and MSI2 fused with a catalytically dead ADAR (MSI2-ADARcd) was not significantly different¹. Other controls in this paper consisted of MSI2 with RNA-recognition motif (RRM) mutations or full deletion of both RRMs, to render MSI2 unable to bind targets to help determine background levels of editing in addition to the empty vector. Furthermore, even overexpression of an ADAR only construct does not cause a significant increase in editing events as compared to both the MSI2-ADARcd, MSI2 RRM (mut/del) ADAR, and the empty vector. We deemed the empty vector is in equivalent control because the catalytically dead had no significant editing.

We also layered an additional filter by only choosing MSI2-ADAR cells treated with MSI2 inhibitor, Ro, sample set in our manuscript. Doing so allowed us to further narrow our on MSI2 Ro sensitive targets.

To address the comment about some figures still having low resolution, we have made corrections to enhance resolution of figures and checked each individually.

We would like to thank Reviewer #3 for their time and input to improve this manuscript.

1. Nguyen, D.T.T., Lu, Y., Chu, K.L. *et al.* HyperTRIBE uncovers increased MUSASHI-2 RNA binding activity and differential regulation in leukemic stem cells. *Nat Commun* **11**, 2026 (2020). <https://doi.org/10.1038/s41467-020-15814-8>